# Using Optimized Two and Three-Band Spectral Indices and Multivariate Models to Assess Some Water Quality Indicators of Qaroun Lake in Egypt

Salah Elsayed [1,*], Mohamed Gad [2], Mohamed Farouk [3], Ali H. Saleh [4], Hend Hussein [5], Adel H. Elmetwalli [6], Osama Elsherbiny [7], Farahat S. Moghanm [8], Moustapha E. Moustapha [9], Mostafa A. Taher [10,11], Ebrahem M. Eid [10,12] and Magda M. Abou El-Safa [4]

[1] Agricultural Engineering, Evaluation of Natural Resources Department, Environmental Studies and Research Institute, University of Sadat City, Sadat City 32897, Egypt
[2] Hydrogeology, Evaluation of Natural Resources Department, Environmental Studies and Research Institute, University of Sadat City, Sadat City 32897, Egypt; mohamed.gad@esri.usc.edu.eg
[3] Agricultural Engineering, Surveying of Natural Resources in Environmental Systems Department, Environmental Studies and Research Institute, University of Sadat City, Sadat City 32897, Egypt; farouk@esri.usc.edu.eg
[4] Environmental Geology, Surveying of Natural Resources in Environmental Systems Department, Environmental Studies and Research Institute, University of Sadat City, Sadat City 32897, Egypt; ali.saleh@esri.usc.edu.eg (A.H.S.); magda.aboelsafa@esri.usc.edu.eg (M.M.A.E.-S.)
[5] Geology Department, Faculty of Science, Damanhour University, Damanhour 22511, Egypt; hendhussein@sci.dmu.edu.eg
[6] Department of Agricultural Engineering, Faculty of Agriculture, Tanta University, Tanta 31527, Egypt; adel.elmetwali@agr.tanta.edu.eg
[7] Agricultural Engineering Department, Faculty of Agriculture, Mansoura University, Mansoura 35516, Egypt; osama_algazeery@mans.edu.eg
[8] Soil and Water Department, Faculty of Agriculture, Kafrelsheikh University, Kafr El-Sheikh 33516, Egypt; fsaadr@yahoo.ca
[9] Department of Chemistry, College of Science and Humanities, Prince Sattam Bin Abdulaziz University, Al-Kharj 11942, Saudi Arabia; m.moustapha@psau.edu.sa
[10] Biology Department, College of Science, King Khalid University, Abha 61321, Saudi Arabia; mostafataherahm@yahoo.com (M.A.T.); ebrahem.eid@sci.kfs.edu.eg or eeid@kku.edu.sa (E.M.E.)
[11] Botany Department, Faculty of Science, Aswan University, Aswan 81528, Egypt
[12] Botany Department, Faculty of Science, Kafrelsheikh University, Kafr El-Sheikh 33516, Egypt
* Correspondence: salah.emam@esri.usc.edu.eg; Tel.: +20-1090305222

**Abstract:** Standard methods are limited for monitoring and managing water quality indicators (WQIs) in real-time and on a large scale. Consequently, there is an urgent need to use reliable, practical, swift, and cost-effective monitoring tools that can be easily deployed and assist decision makers in assessing key indicators relevant to surface water quality in a comprehensive manner. Surface water samples were collected and evaluated for water quality at 16 distinct sites across the Qaroun Lake in 2018 and 2019. Different WQIs, including total dissolved solids (TDS), transparency, total suspended solids (TSS), chlorophyll-a (Chl-a), and total phosphorus (TP), were tested for aquatic utilization. An integrated approach comprising WQIs, geospatial techniques, hyperspectral reflectance indices (SRIs) (commonly used SRIs, two-band and three-band SRIs (Spectral index calculated from water spectral reflectance of two or three wavelengths)), and partial least square regression (PLSR) models were used to assess the water quality of Qaroun Lake. According to the findings, the water quality attributes are polluted to varying degrees. The majority of commonly used SRIs presented moderately relationship with four WQIs (transparency, TSS, Chl-a, and TP) ($R^2$ = 0.45 to 0.64), while the majority of newly two-band SRIs (NSRIs-2b) indicated moderate to strong relationships with WQIs ($R^2$ = 0.51 to 0.74), and the majority of newly three band SRIs (NSRIs-3b) presented strong relationships with WQIs ($R^2$ = 0.67 to 0.81). Broadly, the highest coefficients of determination were noticed with the NSRIs-3b followed by the NSRIs-2b and then the commonly used SRIs. For example, the NSRIs-3b ($NDSI_{648,712,696}$) had stronger relationships with transparency, TSS, and Chl-a with $R^2$ = 0.77, 0.66, and 0.81, respectively, than other SRIs. In addition, the NSRIs-3b ($NDSI_{620,610,622}$) showed the highest

$R^2$ of 0.73 with TSS. The NSRIs-3b coupling with PLSR predicted the WQIs with satisfactory accuracy in the calibration (reach up $R^2 = 0.85$) and validation (reach up $R^2 = 0.81$) datasets. The overall findings of this research study showed that deriving an optimized NSRIs-3b from spectrum region and combining it with PLSR model could be a practical tool for managing water quality of the Qaroun Lake by accurately, timely, and non-destructively monitoring the WQIs.

**Keywords:** 2D correlograms; 3D correlograms; transparency; physicochemical; total suspended solids; PLSR

## 1. Introduction

In recent years, wastewater management has been regarded as one of the most important environmental and public health challenges affecting urban regions in developing countries [1]. Rapid urbanisation, economic expansion and development, as well as changes in lifestyles and consumption habits have resulted in a significant rise in the volume and diversity of waste that needs to be disposed of in an environmentally responsible manner in recent decades [2,3]. This worrying trend has turned into a major issue that must be addressed in order to improve the country's environmental protection [3]. To address these issues, many developing countries have collaborated with their industrialized counterparts to develop national policies and strategies for reducing waste.

Due to rapid population growth and the associated solid and liquid wastes, aquatic systems face significant limitations worldwide, necessitating better management for aquatic usage [4]. This can only be accomplished if integrated water resource management requires an evaluation of all available water resources, including surface water, groundwater, agriculture, residential drainage, and precipitation [5]. Lakes are critical components of the world's water resources, providing water for drinking, irrigation, and power generation, as well as habitat for numerous plant and animal species [6]. The increased input of wastewater induced by fast commercial, industrial, and agricultural growth without building appropriate water infrastructure and treatment facilities has resulted in lake contamination and eutrophication [7,8]. Monitoring water bodies such as lakes and rivers in terms of water quality is crucial since they are among the main sources of fresh water for various purposes (e.g., drinking, irrigation), and thus conserving water in lakes at a minimum level of pollution would be useful.

Qaroun Lake is significant in terms of history and the environment, and it is considered to be a part of a much larger lake known as "the historic Lake Moreis". It was originally a freshwater lake before changing to a saline water ecosystem. Since before 2500 B.C.E., the lake has received fresh water from the Nile [9]. The development of the El Fayoum region accelerated in the second half of the twentieth century, particularly after the construction of the High Dam in 1961, and the lake's salinization increased as the annual Nile flood's freshwater supply was reduced; this increased salinity caused changes in the aquatic biota [10]. There has been extended interest in the past history of the lake level and salinity of Qaroun Lake. Some researchers conducted archeological studies to investigate past lake levels [9], while others investigated the exposed terraces near the lake [11,12].

Qaroun Lake is one of Egypt's most significant inland aquatic habitats, serving as a natural discharge region for El-Fayoum province [13]. Furthermore, Qaroun Lake is a significant place for fishing, tourism, salt manufacturing, and migrating birds. Therefore, it has been designated as a natural protectorate by Prime Ministerial Decree No. 943/1989, in accordance with Law 102/1983 [14,15], due to the great diversity in biological life, archeological locations, and geologic formations (EEAA/NCS, 2007). Recently, Qaroun Lake has shown evidence of stress due to the consequences of numerous industrialisation and urbanization projects. Several pollution sources are met around the lake's southern edge, including El-Fayoum province's agricultural and urban wastewater discharges, as well as fisheries [15,16]. El-Fayoum province's drainage system consists of three main

drains (El-Bats, El Mashroah, and El-Wadi drains) and a number of minor drains that flow into the lake. The lake receives 450 million m$^3$ of mixed untreated effluent annually contaminated by agrochemicals from El-Fayoum province [14,15,17–19]. The aquatic environment contamination by both inorganic and organic contaminants is a key issue providing a serious danger to the viability of the aquatic system [20]. Because there are no drainage exits, the lake's water is lost only through evaporation [21].

Surface water quality is threatened by both natural (evaporation, rainfall, erosion, etc.) and man-made (industrial and agricultural) activity [22,23]. The phrase "water quality" is commonly used in several previous scientific papers concerning the requirements of sustainable management [24]. Water quality criteria are crucial to the health of all living creatures in an aquatic habitat. As a result of the growing human population and the resulting increase in environmental dangers, lake monitoring and evaluation has become an important element of lake management. Thus, investigating water quality indicators in aquatic environments is an essential to comprehend their influence on water quality and living organisms. Many physical, chemical, and biological markers influence water quality in an aquatic habitat [25]. The physicochemical and biological features of a lake environment may be monitored using a variety of approaches, ranging from simple testing to expert studies [26–30]. For this study, several physicochemical water quality parameters were assessed such as TDS, pH, temperature, transparency, TSS, chl-a, and TP as the major indicators of water quality and important parameters in assessing the water quality for aquatic utilization. The monitoring and assessment of water quality indicators are required not only to assess the impact of various sources of pollution but also to preserve aquatic life and develop efficient water resource management [31].

By monitoring the physical, chemical, and biotic properties of lakes, temporal changes can be detected easily [32]. The water quality indicators can be assessed using new technique as the Geographic Information System (GIS) supported by statistical modeling to diminish the cost and time in addition to increasing accuracy [33]. The real world can be represented by GIS through integrated layer of constituent spatial information [34]. Ground based-remote sensing based on spectral reflectance has been commonly used to evaluate surface water quality [35], which provides the spatial information that is not easily available from field campaigns, facilitating the assessment of landscape characteristics [36,37]. The standard methods for monitoring and managing the WQIs in real-time and on a large scale are limited, and thus there is an urgent need to use reliable, practical, fast, and cost-effective monitoring tools that can be easily deployed and assist decision makers in assessing key indicators relevant to water quality in a comprehensive manner [38–41]. To overcome this limitation, the WQIs can be estimated using remote sensing measurements. Different airborne, satellite, or proximate remote sensing approaches have been proven to be cost-effective and usable on a large scale for the integrative assessment of several water quality indicators as a result of rapid improvements in space information and increased utilization of computer applications [42–45]. The optical characteristics of the water surface are inextricably linked to changes in the water's physical, chemical, and biological aspects. As a result, the spectral signatures reflected from the water surface can be used to assess, directly or indirectly, various WQIs, such as TSS, TP and Chl-a, and ammonia nitrogen (NH3-N). Vinciková et al. [46] found that the spectral index including two band at 714 nm and at 650 nm presented the best determine for Chl concentrations and the spectral reflectance at 806 nm is good indicator to estimate TSS. Abd-Elrahman et al. [47] found that a strong relationship with two-band and three-band spectral indices calculated from hyperspectral imaging reflectance with Chl-a. Maliki et al. [48] found that salinity index 2 (SI-2) derived from the green and blue bands of Landsat could be used to assess TDS. Furthermore, changes in the TP concentrations in water can be detected by spectral bands in the blue (450–510 nm) and green (500–600 nm) regions [44,49–51].

Spectral reflectance indices (SRIs) often show inconsistency in estimating the WQIs under different environmental and spatial conditions; thus, it is still necessary to develop further optimized SRIs in order to ensure the performance of SRIs as a simple and rapid

approach to accurately estimate water quality indicators. Generally, the majority of previous studies have focused on the use of two-band spectral indices and few studies have focused on using three-band spectral indices to assess water quality indicators of surface water. An advantage of this study is that the optimized two band (2b) and three-band (3b) spectral indices were selected by establishing 2D and 3D correlogram maps, which offers a high ability to optimally select the spectral indices.

Because spectral measurements generate a huge data set, analyzing spectral reflectance data with an appropriate statistical model remains an important step toward determining the optimum relationship between spectral data and different WQIs. For that, in addition to the derivation of algorithms formulated using individual bands or band ratios, the use of multivariate models using the partial least square regression (PLSR) based on several spectral bands or spectral reflectance indices was as an effective approach to estimate the various water quality indicators [4,52,53]. PLSR models can enable the efficiency of multivariate algorithms (having greater SRIs data) to be compared with more conventional band ratio approaches to algorithm formulation [54]. Moreover, the PLSR is mainly powerful in cases of spectral analysis when the number of predictor variables (i.e., wavelengths or SRIs) is massively greater than the number of observations (i.e., Chl-a). Multivariate integration methods such as PLSR have been proposed to resolve the strong multi-collinear and noisy variables in visible (VIS) and short-wave infrared (SWIR) spectrum data and to efficiently assess the water quality indicators [55]. In this regard, PLSR may offer a deep insight into the potential effectiveness of spectral un-mixing approaches.

There is little information available to assess the advantages of PLSR models based on different types of SRIs (commonly used SRIs, NSRIs-2b and NSRIs-3b) to predict WQIs (TDS, Transparency, TSS, Chl-a, and TP). Therefore, the primary purposes of this study were as follows: (i) to evaluate the drift in physicochemical water quality indicators of the Qaroun Lake in the context of anthropogenic pressure using integrated approach of field campaign, laboratory analysis, and geospatial techniques; (ii) to establish 2D and 3D correlogram maps to select the optimized two- and three-band SRIs; (iii) to compare the performance of different commonly used SRIs, NSRIs-2b and NSRIs-3b, in estimating the WQIs (TDS, Transparency, TSS, Chl-a and TP); and (iv) to evaluate the performance of these different SRIs coupled with PLSR models in predicting the WQIs.

This research aims to provide tools for making better decisions about Qaroun Lake's water assessment to ensure efficient management, assisting in the identification of pollution sources and providing a better vision for the redesign of sampling strategies by focusing on the most effective water quality parameters.

## 2. Materials and Methods

### 2.1. Study Area

Qaroun Lake is part of the El-Fayoum Depression, which was produced by natural circumstances in the Western Desert section of Egypt. Qaroun Lake is a closed shallow brackish lake with an area of about 200 km$^2$ that lies between longitudes of $30°24'$ and $30°50'$ E and latitudes of $29°24'$ and $29°33'$ N (Figure 1), constituting the lowest portion of the Fayoum Depression with no outflow except evaporation. The research area is rectangular and elongated in design, measuring 45 km long, 5.7 km wide, and 4.2 m deep on average. The urban and agricultural regions border the Lake on the south and east, while the uninhabited desert lands border it on the north and west. Qaroun Lake serves as a large natural reservoir for various effluents (agricultural, domestic, sewage, and industrial wastes) that flow through the eastern and southern drains from a large portion of El Fayoum province [18]. The drainage system has two major drains (El-Bats and El-Wadi) as well as a number of smaller drains (e.g., Sheikh Allam and Bahr Qaroun) that go to the lake.

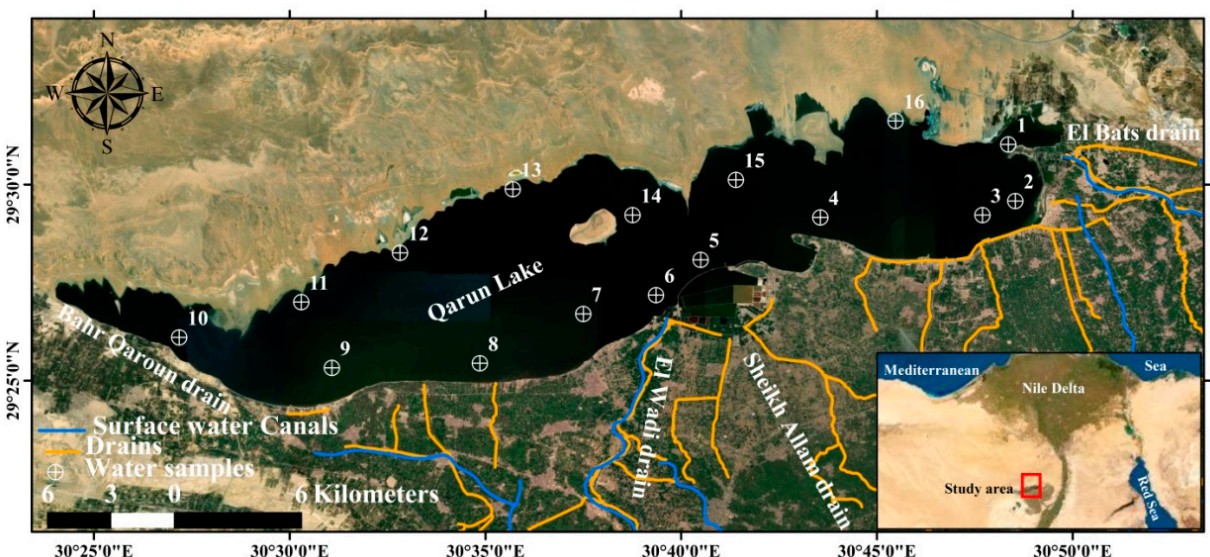

**Figure 1.** Location map of Qaroun Lake and measuring stations.

## 2.2. Sampling and Analyses

During the years 2018 and 2019, surface water samples were collected from 16 locations across the entire lake (Figure 1). The locations of the collected samples were recorded in Universal Transverse Mercator (UTM) using a hand-held MAGELLAN (GPS 315). Lake sampling was conducted in a strategic manner in an aim to collect water samples across the greatest possible water quality gradient. Several physicochemical parameters in water samples were evaluated in situ using a portable calibrated glass electrodes multi-parameter instrument (YSI Professional Plus), including, pH, and temperature (T °C). In addition, a conventional Secchi disc with a diameter of 30 cm was used to measure transparency. For laboratory analysis, water samples were stored in 2-liter polyethylene bottles in an ice box. TDS, TSS, Chl-a, and TP were measured in the obtained samples using standard methods [56–58]. TDS was measured by (GF/C) filtering a certain volume of sample and evaporating at 180 °C. TSS is calculated by subtracting TSS from TDS using glass filter paper. The biomass of phytoplankton (Chl-a) was measured using spectrophotometry [59]. After alkaline persulfate digestion according to standard methods [58], TP was measured using a UV–visible detector and a TP test reagent (TP-HR; C-MAC) based on the breakdown of a phosphorus complex utilizing reactive phosphates. All laboratory chemical analyses were performed at Environmental and Food Lab, University of Sadat City, which accredited according to ISO/IEC 17025/2017, and the precision of the methods were certain by testing certified reference materials (ERM-CA713). Duplicates were performed during the analysis for quality assurance and quality control (QA/QC) of the surface water samples to provide better data confidence from the analytical procedure.

## 2.3. Spatial Distributions of Water Quality Indicators

The ArcGIS Spatial Analyst v.10.2.1 extension includes tools for spatial data analysis that employ statistical theory and methodologies to model spatially referenced data. The interpolation methods in ArcGIS Spatial Analysis were utilized to determine the intervening values for the five determined WQIs (TDS, transparency, TSS, Chl-a, and TP). The maps of five water quality indicators were constructed with the help of the GIS method using inverse distance weighted interpolation (IDW) technique, which is considered one of the simplest and most often used interpolation methods for mapping various parameters. Depending on an estimate, the value at an unsampled location can be estimated as a weighted average of values at points within a particular cut-off distance, or from a specified number of the closest points (typically 10 to 30), where weights are usually inversely proportional

to a power of distance [60,61]. These approaches are useful for processing spatial data at local and global scales, as well as for managing surface water resources [62,63].

### 2.4. Ground-Based Remote-Sensing Measurements

A handheld spectrometer (tec5 AG, Oberursel, Germany) was used to collect radiometric in situ ground-based reflectance measurements from surface water samples. The instrument consists basically of two main units, one of which is connected to a diffuser and measures light radiation as a reference signal, while the other measures spectral reflectance from surface water samples at the spectral range of 302 and 1148 nm and is interpolated to a final spectral resolution of 2 nm. Water samples were put in black cylindrical cups with a diameter of 25 cm and a 10 cm depth, and the spectrometer was held vertically at a nadir position roughly 25 cm above the water surface with a scanning area of 0.05 m$^2$. The reflectance was determined by correcting spectrometer results with a calibration factor obtained from a white reference standard (Apolytetrafluoroethylene white Spectralon reflectance panel). Each surface water sample's spectral reflectance was acquired 3 times, for a total of 15 scans. The measured spectrum for a surface water sample was calculated as the mean of three measurements. To keep fluctuation at a minimum and reduce the impact of changes in sun zenith angle, spectra were taken around noon time. Finally, the spectral reflectance was smoothed to remove noise at both ends of the electromagnetic spectrum.

### 2.5. Selection of Newly Constructed and Commonly Used Spectral Reflectance Indices

Six commonly used indices and sixteen newly derived SRIs were evaluated in this study, as shown in Table 1. The Commonly Used SRIs were chosen based on their sensitivity to changes in water bodies or WQIs. The formula and references of these SRIs are presented in Table 1. The new two-band (NSRIs-2b) and three-band (NSRIs-3b) SRIs were established using 2D and 3D correlogram maps, respectively. Different 2D correlogram maps were established using the lattice package in R statistics ver. 3.0.2 (R Foundation for Statistical Computing, 2013), while 3D correlogram maps were established using MATLAB 7.0 (The MathWorks, Inc., Natick, MA, USA).

These correlogram maps were constructed using all data of two seasons. The 2D maps present the R$^2$ values for the sequential linear regression between WQIs and possible combinations between any two wavelengths in the full spectrum range (302–1148 nm). This range of spectral reflectance is sensitivity to assess the chemical indicators. The 3D correlogram maps present the R$^2$ values for the sequential linear regression between WQIs and possible combinations between any three wavelengths in the visible (VIS) and red-edge from 390 to 750 nm, as shown in.

The NSRIs-2b was calculated as a ratio spectral index according to Gad et al. [55] as shown in Table 1 and as follows:

$$RSI = R_1 / R_2 \tag{1}$$

R$_1$ and R$_2$ are the values of spectral reflectance at selected wavelengths.

Meanwhile, the NSRIs-3b were calculated as a normalized difference index as shown in Table 1 and as follows:

$$NDI = (NIR_3 - NIR_1 - NIR_2)/(NIR_3 + NIR_1 + NIR_2) \tag{2}$$

R$_1$, R$_2$, and R$_2$ are the values of spectral reflectance at selected wavelengths.

**Table 1.** Description of different spectral indices tested in this study.

| SRIs No. | Spectral Reflectance Indices | Formula | References |
|---|---|---|---|
| | **Commonly used SRIs** | | |
| SRI-1 | Ratio spectral index ($RSI_{440,550}$) | $R_{440}/R_{550}$ | [64] |
| SRI-2 | Ratio spectral index ($RSI_{700,670}$) | $R_{700}/R_{670}$ | [65] |
| SRI-3 | Ratio spectral index ($RSI_{806,571}$) | $R_{806}/R_{571}$ | [66] |
| SRI-4 | Ratio spectral index ($RSI_{714,650}$) | $R_{714}/R_{650}$ | [67] |
| SRI-5 | Ratio spectral index ($RSI_{850,550}$) | $R_{850}/R_{550}$ | [68] |
| SRI-6 | Green normalized difference vegetation index (GNDVI) | $(NIR - Green)/(NIR - Green)$ | [69] |
| | **NSRIs-2b** | | |
| SRI-7 | Ratio spectral index ($RSI_{620,608}$) | $R_{620}/R_{608}$ | This work |
| SRI-8 | Ratio spectral index ($RSI_{688,648}$) | $R_{688}/R_{648}$ | This work |
| SRI-9 | Ratio spectral index ($RSI_{700,650}$) | $R_{700}/R_{650}$ | This work |
| SRI-10 | Ratio spectral index ($RSI_{670,470}$) | $R_{670}/R_{470}$ | This work |
| SRI-11 | Ratio spectral index ($RSI_{1130,470}$) | $R_{1130}/R_{470}$ | This work |
| SRI-12 | Ratio spectral index ($RSI_{1130,488}$) | $R_{1130}/R_{480}$ | This work |
| | **NSRIs-3b** | | |
| SRI-13 | Normalized difference spectral index ($NDSI_{648,712,696}$) | $(R_{648}-R_{712}-R_{696})/(R_{648}+R_{712}+R_{696})$ | This work |
| SRI-14 | Normalized difference spectral index ($NDSI_{694,646,710}$) | $(R_{694}-R_{646}-R_{710})/(R_{694}+R_{646}+R_{710})$ | This work |
| SRI-15 | Normalized difference spectral index ($NDSI_{618,646,488}$) | $(R_{618}-R_{646}-R_{448})/(R_{618}+R_{646}+R_{448})$ | This work |
| SRI-16 | Normalized difference spectral index ($NDSI_{618,646,490}$) | $(R_{618}-R_{646}-R_{490})/(R_{618}+R_{646}+R_{490})$ | This work |
| SRI-17 | Normalized difference spectral index ($NDSI_{610,614,608}$) | $(R_{610}-R_{614}-R_{608})/(R_{610}+R_{614}+R_{608})$ | This work |
| SRI-18 | Normalized difference spectral index ($NDSI_{620,610,622}$) | $(R_{620}-R_{610}-R_{622})/(R_{620}+R_{610}+R_{622})$ | This work |
| SRI-19 | Normalized difference spectral index ($NDSI_{696,650,712}$) | $(R_{696}-R_{650}-R_{712})/(R_{696}+R_{650}+R_{712})$ | This work |
| SRI-20 | Normalized difference spectral index ($NDSI_{696,712,648}$) | $(R_{696}-R_{712}-R_{648})/(R_{696}+R_{712}+R_{648})$ | This work |
| SRI-21 | Normalized difference spectral index ($NDSI_{588,576,598}$) | $(R_{588}-R_{576}-R_{598})/(R_{588}+R_{576}+R_{598})$ | This work |
| SRI-22 | Normalized difference spectral index ($NDSI_{618,646,526}$) | $(R_{618}-R_{646}-R_{526})/(R_{618}+R_{646}+R_{526})$ | This work |

### 2.6. Partial Least Squares Regression (PLSR)

The PLSR is a multivariate statistical analysis method that is useful in chemometrics [70]. It is a useful technique for dealing with data when the number of input variables is substantially more than the number of output variables, and collinearity and noise in the data of input variables are high. The PLSR models were formulated using the spectra measurements collected from water of the Qaroun Lake at various stations as predictor variables and the contemporaneous WQIs as the single response variable. In this research, PLSR was used with leave-one-out cross-validation (LOOCV) to link the input variables (SRIs of each group indicated in Table 1) to the output variables (water quality parameters). The number of latent factors (ONLFs) was determined using (LOOCV), and the best ONLFs are that yielding the greatest $R^2$ and the smallest root mean square error (RMSE) in order to represent the calibration data with no over-fitting or under-fitting. The datasets were subjected to random 10-fold cross-validation to improve the results' robustness, as suggested by the software program (Unscrambler X software Version 10.2). (CAMO Software AS, Oslo). The significance of all relationships was tested by $R^2$ at a significance level of $p \leq 0.01$ and 0.001.

### 2.7. Data Analysis

The results of physicochemical water quality parameters in the collected water samples from Qaroun Lake in the two years were statistically analyzed to determine varying statistical parameters (e.g., minimum, maximum, mean, and standard deviation) of the WQIs. The significant differences between the mean values of TDS, transparency, TSS, Chl-a, and TP, and different types of SRIs among 16 stations were compared using Duncan's test at a $p \leq 0.05$ significance level. The relationships between the five water quality indicators and different types of SRIs were examined across two years using a simple linear regression. This statistical analysis was run using SPSS package (v. 12.0, SPSS Inc., Chicago, IL, USA).

## 3. Results and Discussion

### 3.1. Water Quality Indicators and Spatial Distribution Maps

The WQIs of Qaroun Lake were investigated during a two-year period, and different water quality criteria are used to assess water quality. Table 2 shows the mean values, standard deviations, and ranges of the chemical analysis results of the collected samples across two years.

**Table 2.** Statistical description (minimum (min), maximum (max), mean, and standard deviation (SD)) of water quality indicators in Qaroun Lake across two years.

| | **Water Quality Indicators** | | | | | | |
|---|---|---|---|---|---|---|---|
| | **TDS** | **pH** | **Temp.** | **Transparency** | **TSS** | **Chl-a** | **TP** |
| | **First year 2018 (n = 16)** | | | | | | |
| **Min** | 27,704.74 | 7.70 | 28.80 | 30.00 | 12.64 | 0.012 | 0.1147 |
| **Max** | 38,797.87 | 8.30 | 32.30 | 125.00 | 53.72 | 0.146 | 0.5947 |
| **Mean** | 35,616.34 | 8.09 | 30.94 | 70.00 | 35.39 | 0.086 | 0.3423 |
| **SD** | 2627.97 | 0.14 | 0.85 | 31.03 | 16.54 | 0.049 | 0.1995 |
| | **Second year 2019 (n = 16)** | | | | | | |
| **Min** | 27,704.74 | 7.70 | 28.80 | 30.00 | 12.64 | 0.012 | 0.1175 |
| **Max** | 38,797.87 | 8.30 | 32.30 | 125.00 | 53.72 | 0.146 | 0.6453 |
| **Mean** | 35,616.34 | 8.09 | 30.94 | 70.00 | 35.39 | 0.086 | 0.3601 |
| **SD** | 2627.97 | 0.14 | 0.85 | 31.03 | 16.54 | 0.049 | 0.2194 |
| | **Data across two years (n = 32)** | | | | | | |
| **Min** | 27,652.27 | 7.70 | 28.80 | 25.00 | 11.21 | 0.012 | 0.1147 |
| **Max** | 39,056.09 | 8.40 | 34.20 | 125.00 | 62.34 | 0.166 | 0.6453 |
| **Mean** | 35,679.37 | 8.16 | 31.15 | 65.93 | 36.41 | 0.091 | 0.3513 |
| **SD** | 2500.70 | 0.15 | 1.02 | 29.85 | 17.16 | 0.050 | 0.2065 |

All water quality parameters are expressed in mg/L except EC (ms/cm), pH, Temperature (T °C), and Transparency (cm).

The measured water temperature varied between a minimum of 28.8 °C to a maximum of 34.2 °C, with an annual average value of 31.5 °C during the summer (Table 2). Although water in Qaroun Lake lies in the optimal range for most aquatic organisms, the steep temperature gradients can have a remarkable and direct negative impact on fish according to the Canadian Council of Ministers of the Environment (CCME [71]) for aquatic utilization. Weak alkaline water samples were noticed in Qaroun Lake, where pH ranged from 7.7 to 8.3 with the mean of 8.0 pH values (Table 2). Fluctuations of pH levels could be the result of the phytoplankton's photosynthetic status, since pH value is strongly regulated by the rate of $CO_2$ consumption through the photosynthetic activities of phytoplankton [72].

For protecting aquatic life, the CCME [71] recommended that pH values should be categorized in the range of 6.5–9.0, which is acceptable in the water of Qaroun Lake. The TDS values showed that a significant variation from station 16 to station 1 with the respective maximum and minimum values of 38,842 and 28,246 mg/L across two years, demonstrated remarkable significant changes at varying water sampling stations (Table 3) and thus a remarkable spatial variation ranging from lowest levels noticed at the eastern part in front of El-Bats drain to the highest levels recorded at the western part (Figures 2a and 3a). Qaroun Lake's water at the eastern portion is significantly affected by the fresh water of El-Bats drain directly discharged into this area, which moves from east to west, where salinity increases. One of Qaroun Lake's environmental issues is increasing salinity, which is caused by the lake's high evaporation rate and a massive amount of discharged wastewater. According to Sugie et al. [73], excessive salinity has a negative impact on the metabolic activity of phytoplankton, which affects the entire food chain. The spatial distribution map of salinity showed that TDS generally increases from the east to west direction (Figures 2a and 3a).

**Table 3.** Variation of Water Quality Indicators of surface water in Qaroun Lake averaged over two years.

| Stations No. | TDS | Transparency | TSS | Chl-a | TP |
|---|---|---|---|---|---|
| 1 | 28,246g | 37.5fg | 51.99ab | 0.141ab | 0.451c |
| 2 | 34,283f | 42.5e–h | 53.20ab | 0.147ab | 0.501bc |
| 3 | 34,471ef | 47.5e–h | 49.59ab | 0.133ab | 0.620a |
| 4 | 34,495ef | 32.5gh | 45.72b | 0.145ab | 0.530b |
| 5 | 34,500ef | 27.5h | 55.98a | 0.152a | 0.505bc |
| 6 | 34,590ef | 40e–g | 55.465a | 0.141ab | 0.604a |
| 7 | 34,938de | 50d–g | 47.67b | 0.122b | 0.612a |
| 8 | 35,261d | 52.5d–f | 46.51b | 0.121b | 0.529b |
| 9 | 36,762c | 57.5de | 46.28b | 0.085c | 0.322d |
| 10 | 36,801c | 85c | 22.43d | 0.045de | 0.154e |
| 11 | 36,804c | 67.5d | 35.42c | 0.067cd | 0.163e |
| 12 | 36,899c | 112.5ab | 13.43e | 0.024ef | 0.136e |
| 13 | 36,914c | 92.5c | 17.23de | 0.036ef | 0.128e |
| 14 | 38,256ab | 92.5c | 14.91e | 0.053de | 0.120e |
| 15 | 38,806a | 97.5bc | 14.76e | 0.039d-f | 0.116e |
| 16 | 38,842a | 120a | 11.93e | 0.015f | 0.128e |

The same letters are not significantly different from one another based on Duncan's test at a $p \leq 0.05$ significance level.

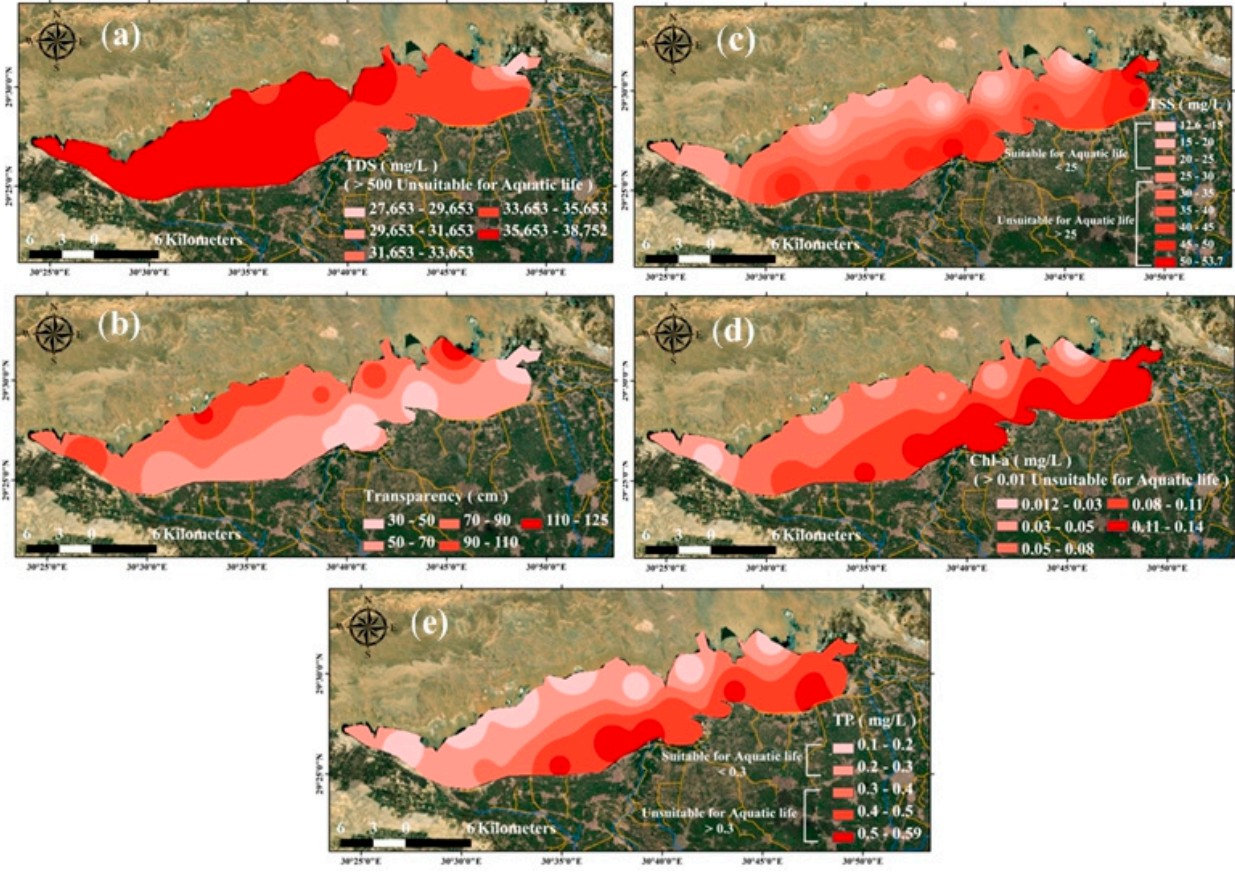

**Figure 2.** Spatial distribution maps of (**a**) total dissolved Solids (TDS), (**b**) transparency, (**c**) total suspended solids (TSS), (**d**) chlorophyll a (Chl-a), and (**e**) total phosphorus (TP) in 2018.

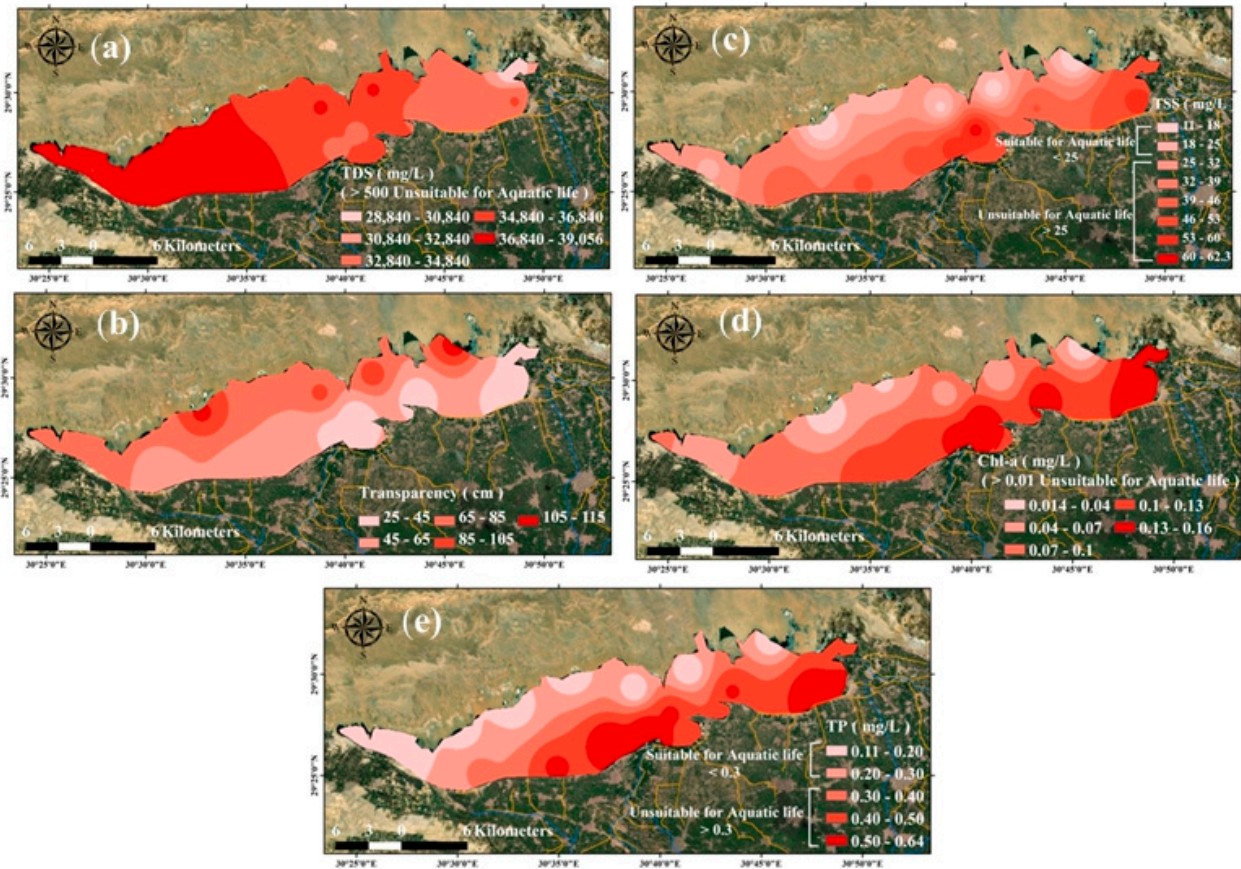

**Figure 3.** Spatial distribution maps of (**a**) total dissolved solids (TDS), (**b**) transparency, (**c**) total suspended solids (TSS), (**d**) chlorophyll a (Chl-a), and (**e**) total phosphorus (TP) in 2019.

The lake's transparency exhibited the lowest values at stations 6 and 1 (30 and 35 cm, respectively), while the highest value was recorded at stations 12 and 16 (115 and 125 cm, respectively) in Table 3. Although the maximum value of transparency was found at station 16, no specific trend for transparency values was observed from station 1 to station 16 (Figures 2b and 3b). The stations in front of the El-Bats and El-Wadi drains exhibited the lowest transparency records. In addition, Fishar et al. [74] verified that the western and upper sections of the lake had the highest levels of transparency. This finding demonstrated the degrading effect of dissolved organic materials released through drains. TSS values varied from 11.2 to 62.34 mg/L, which showed that 37.5 % of samples were suitable and 62.5 % were unsuitable for aquatic life across two years (Table 4). Furthermore, the TSS values showed significant differences between the values collected at different stations, with the respective maximum and minimum values recorded at stations 5 and 16 across the two years (Table 3). High spatial variation from the lower to upper Lake were recorded (Figures 2c and 3c), which reflect the effect of drains on the water quality. Many macrobenthic species were discovered in Lake Qaroun, including *Arthropoda*, *Annelida*, *Mollusca*, and *Coelentrata* [6]. Due to the dominance of Corophium acherusicum, which prefers clayey soil, the largest population density of macrobenthos was recorded in front of the El-Wadi drain [75]. On the contrary, stations in front of El-Bats drain showed the lowest population density, which is confirmed by Fishar et al. [74]. As a result, this finding revealed that the El-Wadi drain discharges a considerable amount of organic matter. On the other hand, the dominance of *Chironomus larvae* (pollution indicator) in front of El-Wadi drain revealed the adverse effect of El-Wadi drain. In addition, the dominance of *Annelids,* especially *Limnodrillus,* in front of the El-Wadi drain is attributed to the pollution and high content of organic matter discharged from the drain [76,77]. The TSS concentrations in the eastern

and southern part of the lake can be high enough to cover aquatic creatures, eggs, and larvae of macro invertebrates. This layer can impede adequate oxygen transport, leading to the death of buried organisms.

**Table 4.** Classification of surface water for suitability to aquatic life in Qaroun Lake according to water quality indicators over two years.

| Water Quality Indicators | Aquatic Life Standard [71] | Water Quality Class | Number of Samples (%) | | |
|---|---|---|---|---|---|
| | | | First Year | Second Year | Across Two Years |
| TDS | <500 | Suitable | 0% | 0% | 0% |
| | >500 | Unsuitable | 16 (100.0%) | 16 (100.0%) | 32 (100.0%) |
| Transparency | - | - | - | - | - |
| | - | - | - | - | - |
| TSS | <25 | Suitable | 6 (37.50%) | 6 (37.50%) | 12 (37.50%) |
| | >25 | Unsuitable | 10 (62.50%) | 10 (62.50%) | 20 (62.50%) |
| Chl-a | <0.01 | Suitable | 0% | 0% | 0% |
| | >0.01 | Unsuitable | 16 (100.0%) | 16 (100.0%) | 32 (100.0%) |
| TP | <0.3 | Suitable | 10 (62.50%) | 9 (56.25%) | 19 (59.3%) |
| | >0.3 | Unsuitable | 6 (37.50%) | 7 (43.75%) | 13(40.7%) |

All water quality parameters are expressed in mg/L except Temperature (T °C), pH, and Transparency (cm). (-) means that the transparency indicators are not used to classify surface water for aquatic life.

The Chl-a values in the water of Qaroun Lake varied from 0.012 to 0.166 mg/L, which reflect unsuitable conditions for an aquatic environment [71]. The Chl-a concentration values demonstrated significant changes across the two years of investigation (Table 3). These findings reflect the eutrophication and algal blooming caused by the nutrient enrichment of water [78], which may impact living organisms in the aquatic system, especially in the southern portions of the Lake in front of drains discharged (Figures 2d and 3d). Although TP is a vital component of a healthy aquatic environment, high concentrations can have a detrimental influence on water bodies. TP values showed that 59.3% of samples were suitable, while 40.7% of samples were unsuitable for aquatic life (Table 4). The spatial distribution map of TP in Qaroun Lake showed an increase in TP concentration levels from the northwest to southeast direction, which recorded maximum levels in stations No. 2, 3, 4, 6, 7, and 8 at the southern edge of the Lake (Figures 2e and 3e). These results indicated that TP can enter water through wastewater discharge or the drainage of agricultural lands, and excessive phosphate concentrations may signal the presence of pollution and are primarily responsible for eutrophic conditions, which cause oxygen shortage with deadly implications for fish and other aquatic species [79]. In addition, greater amounts of total phosphorus are of common concern due to their potential to produce nuisance algal blooms [80].

### 3.2. Variation of Spectral Reflectance Indices of Water Surface in Qaroun Lake

The remote-sensing-based estimation of water quality parameters in lakes requires that responses to environmental factors can be determined through resulting variations in the spectral response. This is fundamentally significant as many previous studies proposed that the spectral-based indices can be constructed to estimate spatiotemporal variations in water quality parameters [46,81–83]. Here, we examine if spectral-based indices could be a robust tool in diagnosing the status of lake ecology in terms of various water quality parameters. This was explored by identifying these parameters at different stations across the Qaroun Lake in two successive years, 2018 and 2019, and then relating them to the spectral response.

In line with the expectations, the change in the physical and chemical constituents of the water caused significant differences in the properties of the light reflected by the water

bodies at varying wavebands of the light spectrum regions, which can be used to evaluate surface water of Qaroun Lake. For examples, the results of quantitative analyses showed that the transparency, TSS, and Chl-a values in Table 4 changed from 27.5 to 120, from 11.93 to 55.98 and 0.015 to 0.147, respectively, followed by changes in the values of $NDSI_{648,712,696}$ from $-0.30$ to $-0.332$ in Table 5. Fortunately, these changes result in significant shifts in the SRIs reflected from the water surface at certain wavelengths across the entire spectrum. Different spectral indices (commonly used, two-band and three-band ratio indices) derived as indicators of various water quality indicators (Table 5). It is obvious that different SRIs demonstrated significant changes across the entire lake (stations from 1 to 16). Broadly, similar to the measured WQIs, the extracted SRIs showed significant changes between various measuring stations. There were clear differences in the SRIs values from stations 1 to 8 against the SRIs values from 9–16, and this is a result of the presence and clear distortions in the water indicators' values between stations 1 to 8 and stations 9 to16. In general, the changes in the values of the majority of the spectral indices follow the changes in the values of the WQIs. Vincikova' et al. [46] and Shafique et al. [49] found that changes in the physical, chemical, and biological aspects of water are inextricably linked to the optical characteristics of the water surface. As a result, spectral signatures reflected from the water surface can be used to evaluate various WQIs, either directly or indirectly. There was a gradual increasing or decreasing in the SRIs values (Table 5) with the gradual change in the values of WQIs (Table 3).

**Table 5.** Variation of different three group types of SRIs for surface water in Qaroun Lake as averaged over two years.

| Station NO. | SRI-1 | SRI-2 | SRI-3 | SRI-4 | SRI-5 | SRI-6 | SRI-7 | SRI-8 | SRI-9 | SRI-10 | SRI-11 |
|---|---|---|---|---|---|---|---|---|---|---|---|
| 1 | 0.755de | 0.976a–c | 0.746a | 0.928ab | 0.580a | −0.307a | 0.999a–e | 0.992a–c | 0.972a–c | 1.129a–c | 1.208a–c |
| 2 | 0.694d | 0.988a | 0.745ab | 0.936a | 0.566ab | −0.312a–c | 1.007a | 0.993ab | 0.979a | 1.222ab | 1.262ab |
| 3 | 0.808b–e | 0.970b–d | 0.709a–e | 0.912b–e | 0.532a–e | −0.326a–c | 1.001a–c | 0.984a–e | 0.962c–e | 1.121a–c | 1.167a–d |
| 4 | 0.742de | 0.980ab | 0.738a–c | 0.937a | 0.572a | −0.313a–c | 1.002a–c | 0.994a | 0.978ab | 1.141a–c | 1.191a–c |
| 5 | 0.723de | 0.971b–d | 0.721a–d | 0.914b–d | 0.542a–e | −0.312a–c | 1.004ab | 0.986a–d | 0.964b–d | 1.230a | 1.277a |
| 6 | 0.796c–e | 0.973a–d | 0.726a–d | 0.918a–c | 0.552a–c | −0.306a | 1.002a–c | 0.984a–e | 0.965b–d | 1.123a–c | 1.152a–d |
| 7 | 0.777de | 0.970b–d | 0.724a–d | 0.910b–e | 0.547a–d | −0.311ab | 1.000a–d | 0.981b–f | 0.960c–f | 1.158a–c | 1.192a–c |
| 8 | 0.827a–e | 0.963b–d | 0.660c–e | 0.903c–f | 0.485b–e | −0.358a–c | 0.996b–f | 0.977d–g | 0.954d–g | 1.005cd | 1.106c–e |
| 9 | 0.884a–e | 0.962cd | 0.680a–e | 0.904c–f | 0.506a–e | −0.334a–c | 0.997b–f | 0.979c–g | 0.955d–g | 1.032b–d | 1.119b–e |
| 10 | 1.023a | 0.957d | 0.631e | 0.894d–f | 0.456e | −0.370c | 0.992d–f | 0.972e–g | 0.946fg | 0.850d | 0.994e |
| 11 | 0.994a–c | 0.964b–d | 0.665b–e | 0.898c–f | 0.484b–e | −0.361a–c | 0.996b–f | 0.972e–g | 0.949e–g | 0.895d | 0.991e |
| 12 | 0.907a–d | 0.969b–d | 0.690a–e | 0.901c–f | 0.508a–e | −0.351a–c | 0.991ef | 0.969fg | 0.950e–g | 0.982cd | 1.025de |
| 13 | 0.899a–d | 0.965b–d | 0.682a–e | 0.904c–f | 0.510a–e | −0.342a–c | 0.995c–f | 0.975d–g | 0.953d–g | 0.991cd | 1.064c–e |
| 14 | 0.977a–c | 0.962cd | 0.653de | 0.896d–f | 0.477c–e | −0.360a–c | 0.989fg | 0.969fg | 0.947fg | 0.882d | 0.974e |
| 15 | 0.991a–c | 0.961cd | 0.642e | 0.893ef | 0.464de | −0.362a–c | 0.990f | 0.968fg | 0.945g | 0.871d | 0.971e |
| 16 | 1.001ab | 0.957d | 0.631e | 0.889f | 0.455e | −0.367bc | 0.988g | 0.967e | 0.943g | 0.860d | 0.987e |

| Station NO. | SRI-12 | SRI-13 | SRI-14 | SRI-15 | SRI-16 | SRI-17 | SRI-18 | SRI-19 | SRI-20 | SRI-21 | SRI-22 |
|---|---|---|---|---|---|---|---|---|---|---|---|
| 1 | 1.177a–c | −0.328ab | −0.328a | −0.306a–c | −0.307a–c | −0.334d–f | −0.333a–c | −0.328ab | −0.329ab | −0.33b–d | −0.321ab |
| 2 | 1.219ab | −0.327a | −0.328a | −0.292a | −0.293a | −0.3337f | −0.332a | −0.327a | −0.327a | −0.332ab | −0.314a |
| 3 | 1.146a–c | −0.329a–c | −0.329ab | −0.306a–c | −0.306a–c | −0.3335ef | −0.333a–c | −0.328a–c | −0.329a–c | −0.332a–c | −0.320ab |
| 4 | 1.157a–c | −0.328ab | −0.328a | −0.303ab | −0.304ab | −0.3335ef | −0.333ab | −0.327ab | −0.328ab | −0.33b–d | −0.320ab |
| 5 | 1.242a | −0.328ab | −0.328a | −0.291a | −0.292a | −0.3335ef | −0.333ab | −0.327ab | −0.329ab | −0.3314a | −0.311a |
| 6 | 1.129a–d | −0.329a–c | −0.329a–c | −0.306a–c | −0.306a–c | −0.333d–f | −0.333ab | −0.329a–d | −0.329a–c | −0.332a–c | −0.319ab |
| 7 | 1.167a–c | −0.329a–d | −0.330a–d | −0.302ab | −0.303ab | −0.333c–f | −0.333a–c | −0.329a–d | −0.329a–d | −0.33b–d | −0.317ab |
| 8 | 1.0842c–f | −0.30b–e | −0.330a–d | −0.32b–d | −0.322b–e | −0.333c–f | −0.33b–d | −0.330b–e | −0.330b–e | −0.333c–f | −0.33b–d |
| 9 | 1.106b–e | −0.330a–e | −0.330a–e | −0.319a–d | −0.319a–d | −0.333b–e | −0.334b–e | −0.329a–e | −0.330a–e | −0.332b–e | −0.327a–c |
| 10 | 0.998ef | −0.332de | −0.332c–e | −0.346d | −0.345de | −0.333a–d | −0.335c–f | −0.331de | −0.332de | −0.3331ef | −0.344cd |
| 11 | 0.990ef | −0.331c–e | −0.331b–e | −0.342d | −0.342de | −0.333c–e | −0.334b–f | −0.331c–e | −0.331c–e | −0.3328ef | −0.342cd |
| 12 | 1.014d–f | −0.332e | −0.332c–e | −0.332c | −0.333c–e | −0.333a–c | −0.335d–f | −0.331e | −0.3317e | −0.3328ef | −0.338c–d |
| 13 | 1.053c–f | −0.331c–e | −0.331b–e | −0.33b–d | −0.326b–e | −0.333a–e | −0.334b–f | −0.331c–e | −0.331c–e | −0.333d–f | −0.333b–d |
| 14 | 0.972f | −0.332e | −0.332de | −0.346d | −0.346de | −0.332ab | −0.335df | −0.331e | −0.332e | −0.3332de | −0.346d |
| 15 | 0.970f | −0.332e | −0.332c–e | −0.346d | −0.346de | −0.333a–c | −0.334d–f | −0.331e | −0.332e | −0.3331ef | −0.346d |
| 16 | 0.988ef | −0.332e | −0.333e | −0.347d | −0.347e | −0.3328a | −0.3354f | −0.331e | −0.332e | −0.3331ef | −0.346d |

The same letters are not significantly different from one another based on Duncan's test at a $p \le 0.05$ significance level. The full names of the abbreviations of SRIs are listed in Table 1.

Therefore, the results suggest that SRIs at certain regions of the electromagnetic spectrum could be efficient for the estimation of WQIs, and thus, monitoring ecosystems in lakes by means of remote-sensing-based indices may provide a robust tool for informed decisions for saving lakes from degradation. Many studies have found that the light reflectance in the VIS, red-edge, and NIR of spectrum regions has a strong relationship with different physiochemical water components in varying water bodies, suggesting that these regions of the spectrum could be utilized to assess WQIs of water surface [38,40,84]. In addition, the water quality of Qaroun Lake is influenced by anthropogenic activities,

agricultural activities, and waste from drains. Therefore, the differences in surface water quality between the different stations presented significant differences in the potential of indirect estimation of their quality through the calculation of SRIs. According to Seyhan et al. [84], the characteristics of spectral signatures collected from the water surface are a function of the biological and chemical properties of water bodies.

### 3.3. Ability of Different SRIs for Indirect Assessment Water Quality Indicators

The new SRIs were extracted based on 2D and 3D correlogram maps established using the two years' pooled data of the spectral reflectance data collected from different surface water sampling (Figures 4 and 5).

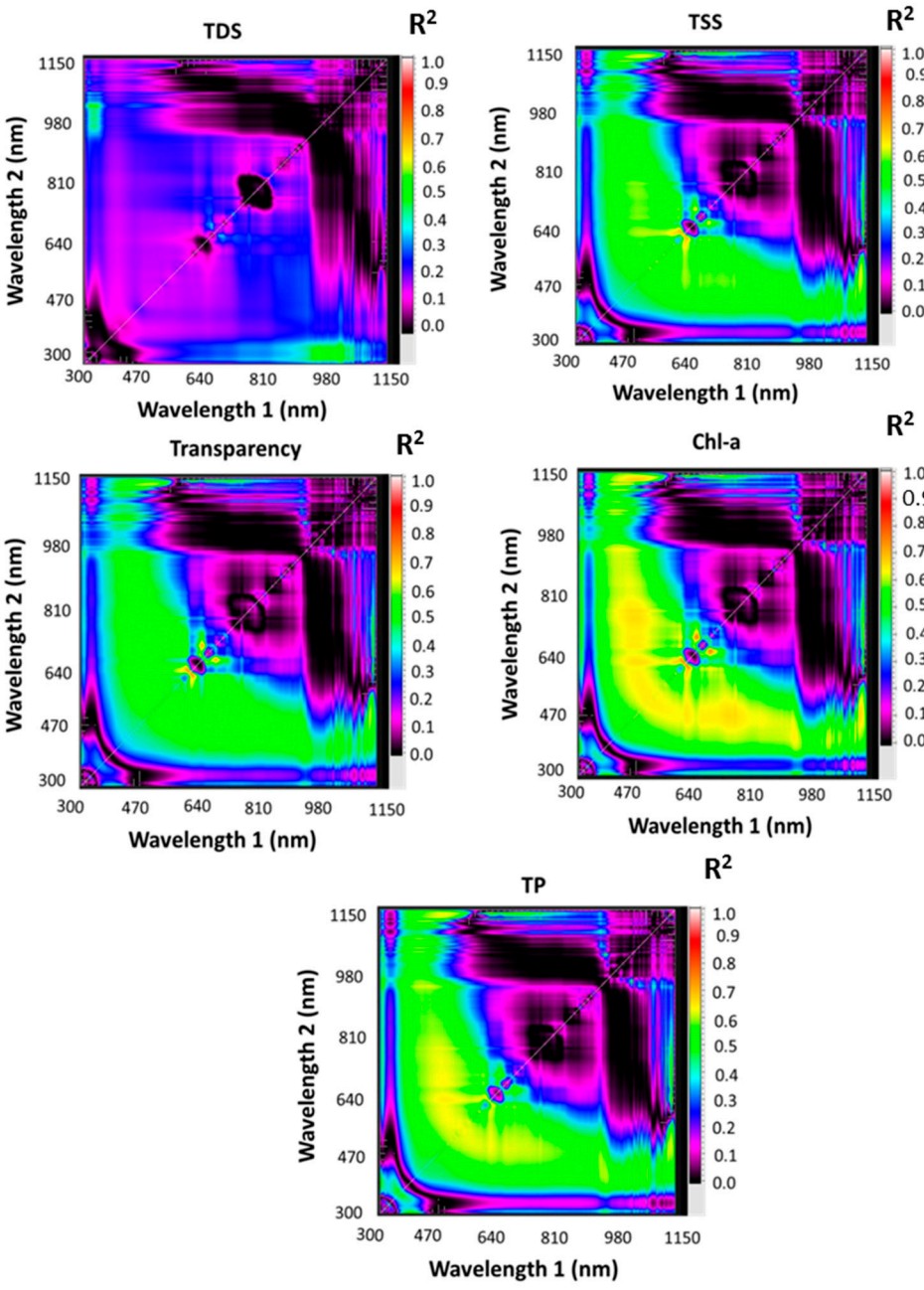

**Figure 4.** Correlation matrices of correlogram maps illustrating the coefficients of determination ($R^2$) for dull wavelength combinations ranged between 302 to 1148 nm of the spectra with total dissolved solids (TDS), transparency, total suspended solids (TSS), chlorophyll a (Chl-a), and total phosphorus (TP) across two years.

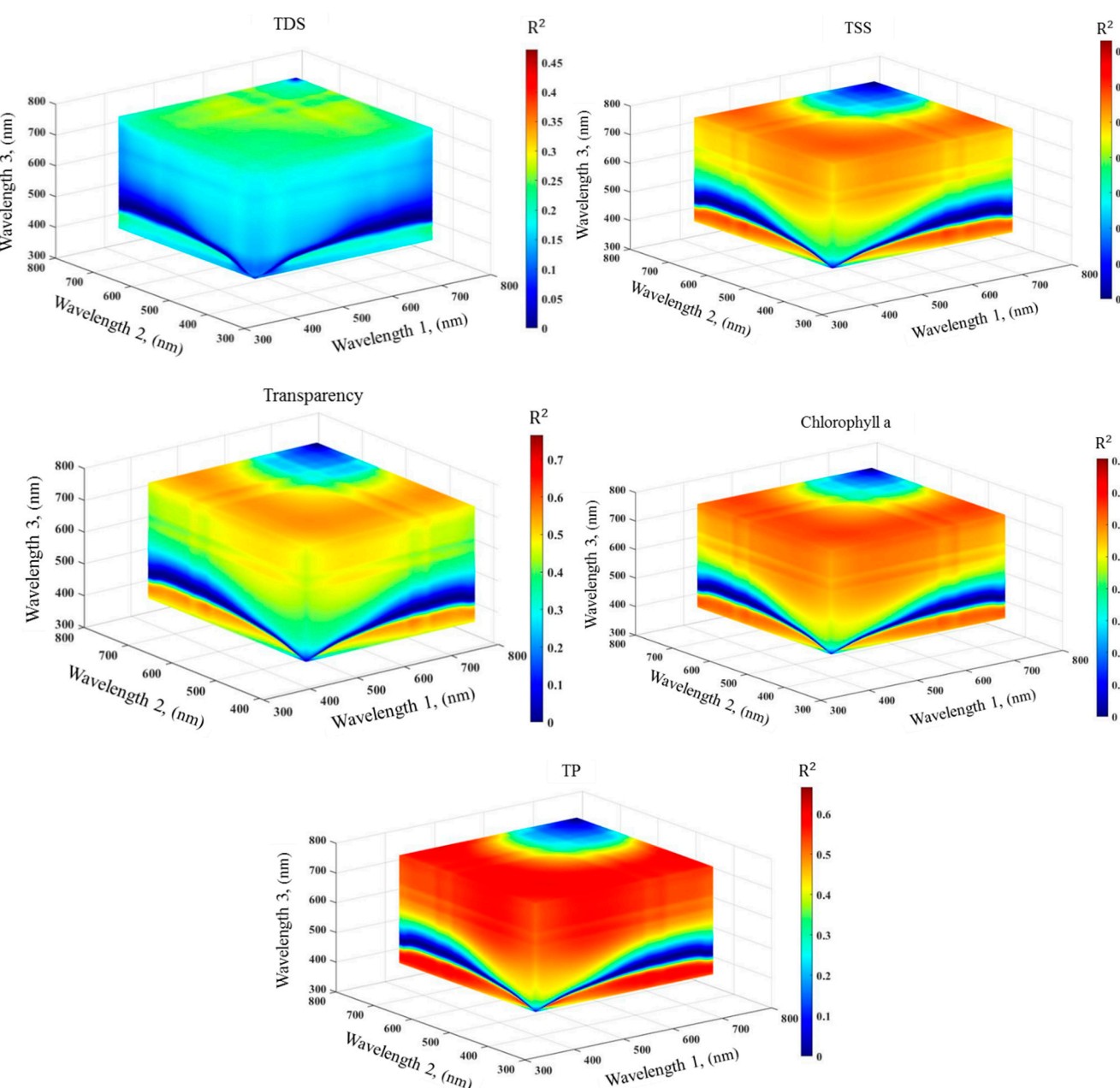

**Figure 5.** Three-dimensional correlogram maps of the coefficients of determination ($R^2$) that were obtained for the relationship between total dissolved solids (TDS), transparency, total suspended solids (TSS), chlorophyll a (Chl-a), and total phosphorus (TP) across two successive years, 2018 and 2019, that was calculated for all possible three-band combinations from 390–750 nm.

These 2D and 3D correlogram maps presented the coefficients of determination ($R^2$) for the relationships between records of WQIs and the SRIs derived from all possible combinations of dual wavelengths of binary in the whole spectral range (302–1148 nm). The hotspot areas based on the colour scale for the best $R^2$ identify the best relationships between the SRIs and WQIs. A hotspot is related to the colour scale, which was used to detect the best $R^2$ between SRI and each WQI. Based on the hotspots (colour scale) of the identified best $R^2$, the commonly used SRIs, NSRIs-2b and NSRIs-3b, were selected based on the combined information from the WQIs in the VIS range (440, 450, 470, 488, 490, 526, 550, 571, 588, 576, 598, 608, 610, 614, 618, 620, 622, 646, 648, 650, 670, 694, and 696 nm), the red-edge range (700, 710, 712, 714, 750, and 806 nm), and in the NIR range (850 and

1130 nm). Then, the NSRIs of the 2D and 3D correlogram maps were selected based on the highest $R^2$.

The coefficient of determination values as indicators for the relationship between various measured water quality indicators and different SRIs (commonly used, newly extracted SRIs; two-band and three-band) are depicted in Figure 6. The most significant relationships for the majority of the algorithms used in combination with TDS ($R^2$ = 0.12–0.32), transparency ($R^2$ = 0.26–0.77), TSS ($R^2$ = 0.28–0.73), Chl-a ($R^2$ = 0.34–0.81), and TP ($R^2$ = 0.27–0.67) were found with linear regression models. Many of the algorithms, whether they use two-band or three-band ratios normalized between the red-edge (620–750 nm) and VIS (400–620 nm) wavelengths, produced comparable results in predicting the WQIs. In general, the majority of the commonly used SRIs presented moderate relationships with four WQIs (transparency, TSS, Chl-a, and TP) ($R^2$ = 0.45 to 0.64), while the majority of NSRIs-2b presented moderate to strong relationships with WQIs ($R^2$ = 0.51 to 0.74), and the majority of NSRIs-3b presented strong relationships with WQIs ($R^2$ = 0.67 to 0.81) in Figure 6.

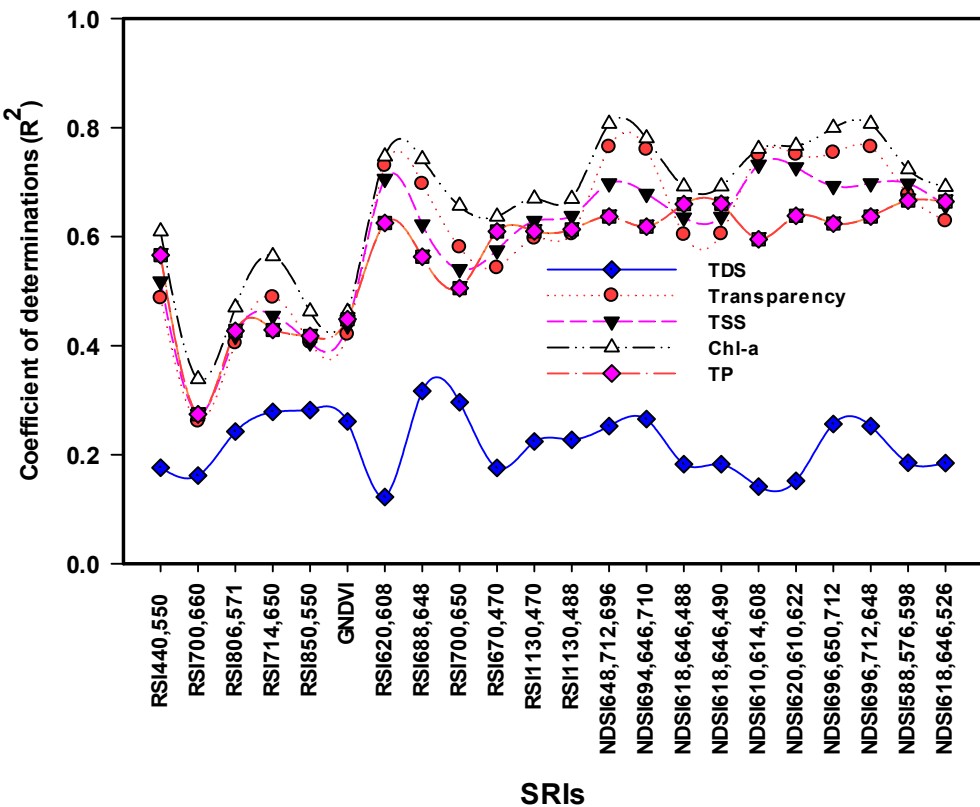

**Figure 6.** Coefficients of determination ($R^2$) for the relationship between various spectral reflectance indices (SRIs) and total dissolved solids (TDS), transparency, total suspended solids (TSS), chlorophyll a (Chl-a), and total phosphorus (TP) across two investigated years.

For Chl-a, the relationship observed between various SRIs and Chl-a was shown to significant in most cases, with the highest coefficient of determination ($R^2$ = 0.81) recorded with the three-band index (NDSI$_{648,712,696}$). Broadly, the highest coefficients of determination were noticed with the NSRIs-3b, followed by NSRIs-2b and then the commonly used SRIs. It seems that the spectral index based on the red-edge regions of the electromagnetic spectrum always produce higher coefficient of determination. This might be due to the red region of spectrum having more sensitivity to the changes in Chl-a. In agreement with our findings, Elhag et al. [81] reported that the maximum chlorophyll index (MCI) constructed from the remote sensing data of Sentinel-2 at wavelengths of 665, 705, and 740 nm from the red-edge regions could be used to estimate the chlorophyll a concentration of water in the dam lake of Wadi Baysh, Saudi Arabia. In addition, MCI presented strong relationship

with chlorophyll a concentration estimated with an $R^2$ of 0.96. Vincikova' et al. [46] found that the spectral index based on 714 and 650 nm constructed from the red-edge regions presented the best estimates for the chl-a concentration of water surface with an $R^2$ of 0.86. In our research, also, the two-band ratio spectral index ($RSI_{700,650}$) showed a good relationship with Chl-a, with an $R^2$ of 0.66. Gitelson [82] investigated the dynamics of the reflectance peak near 700 nm and concluded that it was significant for the remote sensing of inland and coastal waters, particularly for determining chlorophyll concentrations with an $R^2$ of 0.93. Han and Jordan [83] mentioned that the spectral ranges at 630–645 nm, 660–670 nm, 680–687 nm, and 700–735 nm were found to be possible regions where the first derivatives can be employed to estimate Chl concentration. The $R^2$ values reached 0.74 for the wavelength at 686.7 nm. In this study, the 2D and 3D correlograms were established to select the best SRI, which fit well with Chl-a to compare with published SRIs. Abd-Elrahman et al. [47] found that a strong relationship with two-band and three-band spectral indices calculated from hyperspectral imaging reflectance and $R^2$ values were 0.975 and 0.982 for the two- and three-band models, respectively.

Similar to Chl-a, the NSRIs-3b demonstrated strong relationships with TSS with the highest $R^2$ (0.73) recorded with the $NDSI_{620,610,622}$. Vincikova' et al. [46] found that the band ratio algorithm using NIR and red ($R_{806}/R_{670}$) was strongly related to TSS ($R^2 = 0.86$). However, the band ratio based on NIR and green wavelengths ($R_{850}/R_{550}$) was less significant ($R^2 = 0.54$). We also found in our research that the spectral indices derived from NIR with green (495 to 570) showed moderate relationships with TSS such as $RSI_{850,550}$ ($R^2 = 0.41$) and GNDVI ($R^2 = 0.44$). Similar to the relationship between the previously mentioned WQIs and different SRIs, the coefficient of determination between transparency and all SRIs demonstrated reasonable significant values. The greatest $R^2$ was recorded with the index $NDSI_{648,712,696}$ with an $R^2$ of 0.77.

The TP studies include the measurement of all inorganic, organic, and dissolved forms of phosphorus. Phosphates are among main plant nutrients that help plants and algae to grow more quickly. Total phosphorus has a direct relationship with the Chl-a concentration and is indirectly related to transparency or water clarity, which is mainly estimated by Secchi depth [84]. The TP and three-band index, which was used to assess Chl-a ($NDSI_{648,712,696}$), showed the strongest relationships in comparison to commonly used and NSRIs-2b. The highest $R^2$ of 0.66 was observed with the index $NDSI_{648,712,696}$, suggesting that indices based on red and green regions produced higher correlations. Other studies found that the spectral reflectance in the blue range (450–510 nm) and green range (500–600 nm) wavebands are highly sensitive to changes in the total phosphorus concentrations in water [44,49–51]. In contrast to other investigated WQIs, in general, the TDS presented weak relationships with all types of SRIs, and its $R^2$ varied from 0.12 to 0.32. The highest $R^2$ of 0.32 was produced with the $RSI_{688,648}$, while the minimum $R^2$ of 0.12 was recorded with the index $RSI_{620,608}$. In agreement with our results, Gad et al. [55] found that weak relationships were found between TDS and the spectral indices of groundwater samples from the El Fayoum Depression in the Western Desert derived from UV/VIS, UV/NIR, VIS/VIS, VIS/NIR, and NIR/NIR. The reason for that may be due to the changes in four WQIs (transparency, TSS, Chl-a, and TP) are not well correlated with TDS across the Qaroun Lake. Moreover, the changes in spectral reflectance values mainly based on the water colour and water bodies.

It is obvious from the above-mentioned results that the combination between the VIS and red-edge regions of the electromagnetic spectrum, in particular the NSRIs-3b, often provided the most sensitive and robust quantification of water quality parameters. In general, using NSRIs-3b enhances the estimation of various water quality parameters. The multiple wavelengths algorithms were observed to enhance the accuracy of water quality parameters estimations. Generally, it is difficult to use the spectral reflectance of one wavelength of the spectrum to estimate WQI accurately because such an indicator is sensitive to complex factors such as environmental conditions, timeliness, and regional specificity. Moreover, the sensitivity of this wavelength is not constant to assess the changes

in water characteristics concentration under different conditions. However, the NSRIs-3b combining the wavelengths from the different regions of the spectrum displays less saturation. This may explain why the SRIs based on three bands were more accurate at estimating WQIs than the other SRIs based on one or two bands. Wang et al. [85] reported that detecting TSS in moderately clear water using an individual waveband or two waveband combinations is challenging. However, a combination of three wavebands in turbid water bodies was successful in estimating the Chl content.

### 3.4. Performance of PLSR Models to Predict Water Quality Indicators

Although SRIs are a simple tool and several indices have been efficiently used in estimating WQIs, they are hindered by their use of just a few bands and are impacted by environmental conditions, timeliness, and regional specificity [55,81,86]. Furthermore, using several wavebands sensitive to water quality parameters through SRIs combined with PLSR models could enhance the performance of the models to predict the WQIs. Therefore, this study has considered different PLSR models that are based on multiple SRIs of three different groups for improving the estimation of different water quality indicators.

In this research study, the three SRIs groups were applied to the PLSR in order to predict the TDS, transparency, TSS, chl-a, and TP. Most importantly, the PLSR models coupled with NSRIs-3b had the best performances in the estimation of the WQIs in both the calibration and validation datasets, followed by the PLSR models coupled with NSRIs-2b, and then by the PLSR models coupled with commonly used SRIs. Again, the reason for this is that NSRIs-3b combined the wavelengths from the different regions of the spectrum, which are sensitive to changes in WQIs, and they display less saturation effects under environment conditions, timeliness, and regional specificity. This may explain why the NSRIs-3b based on three-bands combined with PLSR models has a higher performance in the prediction of WQIs than the commonly used SRIs and the NSRIs-2b. For example, the calibrated models of the PLSR showed the highest performance to predict the four tested water quality parameters based on the NSRIs-3b with ($R^2_{cal}$ = 0.82, RMSEc = 13.23) for transparency, ($R^2_{cal}$ = 0.78, RMSEc = 8.05) for TSS, ($R^2_{cal}$ = 0.85, RMSEc = 0.02) for Chl-a, and ($R^2_{cal}$ = 0.78, RMSEc = 2.24) for TP (Table 6). In addition, the predictive models of PLSR showed the highest performance to predict the four water quality parameters based on the NSRIs-3b with ($R^2_{val}$ = 0.78, RMSE = 15.69) for transparency, ($R^2_{val}$ = 0.76, MSEv = 8.46) for TSS, ($R^2_{val}$ = 0.81, MSEv = 0.02) for Chl-a, and ($R^2_{val}$ = 0.72, RMSEv = 0.11) for TP (Table 6). The PLSR based on the three different groups of SRIs individually demonstrated the highest performance to predict Chl-a than the other four measured parameters in the calibration and validation models, with $R^2$ ranging from 0.79 to 0.85. The predictive model of the PLSR based on the three different groups of SRIs individually showed the lowest performance to predict the TDS of surface water.

In terms of estimating water quality indicators, the PLSR models outperformed the individual SRIs. This is due to the fact that the various tested PLSR models comprise multiple sensitive wavebands covering all of the main variations in water components and are closely linked to the major changes in the targeted water quality parameters. In accordance with these findings, Wang et al. [85] found that PLSR models based on many selected wavebands were more precise in predicting inland water quality indicators than models based on single- or two-band combinations. The $R^2$ increased from 0.43 and 0.40 for single-band and two-band combinations to 0.98 and 0.97 for PLSR models, respectively, and for Chl-a and TSS prediction using PLSR models based on waveband selection, it increased from 400 to 900 nm. Gad et al. [55] found that the SRI-based PLSR models also provided a clear relationship between calculated and predicted values for all six irrigation water quality indices IWQI parameters. In addition, the sensitive spectral intervals of wastewater for reach of the six water quality parameters combined with extreme learning machine (ELM) and PLSR, namely, chemical oxygen demand (COD), biological oxygen demand (BOD), NH 3 -N, TDS, total hardness (TH), and total alkalinity (TA), were selected using three different methods: gray correlation (GC), variable importance in projection (VIP), and

set pair analysis (SPA). On the whole, the PLSR and ELM both achieved satisfying model accuracy, but the prediction accuracy of the latter was higher than the former, and the R2 of both models varied from 0.79 to 0.98 for the validation of water quality parameters based on the best mode (COD with GC-PLSR model, BOD with GC-ELM model, NH 3 -N with GC-ELM model, TDS with SPA-ELM model, TA with SPA-ELM model, and TH with SPA-ELM model) [45]. Our study focused on applying the PLSR models based on different SRIs groups specially the group of NSRIs-3b, which includes three bands from different regions whose sensitivity to change in water bodies enhanced the prediction of WQIs.

**Table 6.** Results of calibration (equation, $R^2_{cal}$ and $RMSE_C$) and 10-fold cross-validation (equation, $R^2_{val}$ and $RMSE_V$) Partial least squares regression models of the association between three spectral reflectance index types and total dissolved solids (TDS), transparency, total suspended solids (TSS), chlorophyll a (Chl-a), and total phosphorus (TP).

| SRIs types | ONLFs | Measured Variables | Calibration | | | Validation | | |
|---|---|---|---|---|---|---|---|---|
| | | | Equation | $R^2_{cal}$ | $RMSE_C$ | Equation | $R^2_{val}$ | $RMSE_V$ |
| Commonly used SRIs | 1 | TDS | y = 0.2267x + 2759 | 0.23 ** | 2164.49 | y = 0.1765x + 2940 | 0.21 ** | 2312.28 |
| | 3 | Transparency | y = 0.6203x + 25.039 | 0.62 *** | 18.11 | y = 0.5458x + 29.617 | 0.49 *** | 21.75 |
| | 3 | TSS | y = 0.6063x + 14.331 | 0.61 *** | 10.60 | y = 0.5616x + 16.894 | 0.55 *** | 12.24 |
| | 6 | Chl-a | y = 0.7847x + 0.020 | 0.79 *** | 0.02 | y = 0.7847x + 0.020 | 0.73 *** | 0.03 |
| | 3 | TP | y = 0.6311x + 0.130 | 0.63 *** | 0.12 | y = 0.6x + 0.143 | 0.55 *** | 0.14 |
| NSRIs-2b | 5 | TDS | y = 0.4756x + 1871 | 0.47 *** | 1782.39 | y = 0.3446x + 233 | 0.25 ** | 2078.55 |
| | 3 | Transparency | y = 0.7218x + 18.657 | 0.75 *** | 15.11 | y = 0.6658x + 21.062 | 0.69 *** | 17.78 |
| | 4 | TSS | y = 0.7254x + 9.998 | 0.73 *** | 8.85 | y = 0.6731x + 11.749 | 0.60 *** | 10.76 |
| | 6 | Chl-a | y = 0.8153x + 0.017 | 0.82 *** | 0.02 | y = 0.7692x + 0.021 | 0.76 *** | 0.03 |
| | 3 | TP | y = 0.6892x + 0.106 | 0.73 *** | 0.10 | y = 0.6781x + 0.120 | 0.66 *** | 0.12 |
| NSRIs-3b | 2 | TDS | y = 0.2839x + 256 | 0.28 ** | 2012.82 | y = 0.1914x + 28,834 | 0.17 * | 2392.33 |
| | 4 | Transparency | y = 0.7514x + 17.142 | 0.82 *** | 13.23 | y = 0.7109x + 21.13 | 0.78 *** | 15.69 |
| | 4 | TSS | y = 0.7464x + 9.731 | 0.78 *** | 8.05 | y = 0.7119x + 9.957 | 0.76 *** | 8.46 |
| | 5 | Chl-a | y = 0.8305x + 0.016 | 0.85 *** | 0.02 | y = 0.8161x + 0.016 | 0.81 *** | 0.02 |
| | 3 | TP | y = 0.7189x + 0.086 | 0.75 *** | 0.10 | y = 0.704x + 0.091 | 0.72 *** | 0.11 |

Levels of significance: *: $p < 0.05$, **: $p < 0.01$, and ***: $p < 0.001$.

Once again, our results confirm that the PLSR models based on several SRIs can improve the estimation of several WQIs and can be used as a unified technique for the remote quantification of constituent concentrations in water quality evaluation.

*3.5. Outcomes and Practical Applications of the Research*

Water bodies (e.g., lakes and rivers) are important socioeconomic and ecological natural resources. The water of the Qaroun Lake has become more saline over the time, which affects all living creatures across the lake and even the wider public since it is among the area's freshwater resources. With the expected global warming, the water salinity of the lake will increase as a result of massive evaporation in that hot region. Detecting changes in the lake ecosystem is crucial to conserve life across the lake, which needs more advanced techniques. The UN SDGs (sustainable development goals) have placed strict concerns on deriving new and cost-effective strategies for the monitoring of the ecological status of lakes. In this regard, it is suggested that the spatially resolving, regional scale, and data provided by remote sensing techniques would be effective in the operational monitoring of lakes. In our research, characterizing spatial variability at lake-scale has been proven using ground based remotely sensed data which can enable point-sampling measurements to be extrapolated to the wider ecosystem.

A combined approach of ground-based and satellite-based remote sensing can be, therefore, a reliable technique in the assessment and monitoring of ecological status in water bodies (e.g., lakes and rivers). This research has made a contribution to furthering its implication in this regard. The relevance of remote sensing technique to the UN SDGs has been obviously shown, and it has been revealed that remotely sensed data can also provide an important contribution to the understanding of lake functions and processes and thus will indirectly be useful to the wider public. The accurate estimation of water quality parameters was obviously shown achievable using in situ ground-based remote

sensing data. The results further demonstrated that remote sensing can be effectively used to map distribution patterns of various water quality parameters. In an economical point of view, the combined approach of satellite- and ground-based data will be far cheaper in comparison to sample-point measurements. Moreover, manufacturing a simple and cost-effective three-band spectral instrument can help in the primary detection of changes in lakes ecosystems.

## 4. Conclusions

Surface water samples were collected and evaluated for water quality at 16 distinct sites across the Qaroun Lake in 2018 and 2019. The TDS, transparency, TSS, Chl-a, and TP were among the physicochemical water quality indicators (WQIs) that were tested for aquatic utilization in this research. The distribution patterns of five WQIs using GIS maps indicated that the water quality attributes are polluted to varying degrees, and the progressive increase in salinity accelerates the degradation of the lake's aquatic ecosystem. For examples, The TDS of water ranged from 27,652.27 to 39,056.1 mg/L with a mean of 55,749.02 mg/L, which demonstrated a great spatial variation ranging from lowest levels at the eastern part of the lake in front of El-Bats drain to the highest levels at the western part. The WQIs of Qaroun Lake were assessed by different commonly used SRIs, NSRIs-2b and NSRIs-3b, and PLSR models. The results showed that the majority of NSRIs-3b presented strong relationships with the WQIs. The results further showed that the PLSR algorithm models based on SRIs-3b performed the best in estimating the WQIs in both the calibration and validation datasets, followed by the PLSR algorithm models based on SRIs-2b. For example, the predictive models of the PLSR showed the highest performance to predict the five water quality parameters based on the NSRIs-3b, with ($R^2_{val}$ = 0.78, RMSE = 15.69) for transparency, ($R^2_{val}$ = 0.76, MSEv = 8.46) for TSS, ($R^2_{val}$ = 0.81, MSEv = 0.02) for Chl-a, and ($R^2_{val}$ = 0.72, RMSEv = 0.11) for TP. The PLSR based on the three different groups of SRIs individually showed the highest performance to predict Chl-a compared to the other four measured parameters in the calibration and validation models, with $R^2$ varying from 0.79 to 0.85. Finally, these findings suggest that integrating SRIs-3b with PLSR could provide a reliable and accurate method for estimating WQIs in Qaroun Lake. In the future, the method proposed in this study combining spectral indices algorithms and PLSR models could be evaluated further to improve its stability under various conditions of rivers and lakes.

**Author Contributions:** Conceptualization, S.E.; methodology, S.E., M.F., H.H., A.H.E., A.H.S., M.M.A.E.-S., O.E., F.S.M. and M.G.; software, S.E., M.F., A.H.E., H.H., M.M.A.E.-S., A.H.E., O.E., F.S.M. and M.G.; validation, S.E., M.F., H.H., A.H.E., A.H.E., O.E., F.S.M. and M.G.; formal analysis, S.E., M.F., H.H., A.H.E., A.H.E., M.M.A.E.-S., O.E., F.S.M. and M.G.; investigation, S.E., M.F., H.H., A.H.E., A.H.E., M.M.A.E.-S., O.E., F.S.M. and M.G.; resources, E.M.E.; data curation, S.E., M.F., H.H., A.H.E., A.H.E., O.E., M.M.A.E.-S., F.S.M. and M.G.; writing—original draft preparation, S.E., M.F., H.H., A.H.E., A.H.E., O.E., M.M.A.E.-S., F.S.M. and M.G.; writing—review and editing, M.E.M., M.A.T. and E.M.E.; supervision, E.M.E.; project administration, E.M.E.; funding acquisition, E.M.E. All authors have read and agreed to the published version of the manuscript.

**Funding:** This research was funded by the Deanship of Scientific Research at King Khalid University, grant number GRP. 87/39.

**Institutional Review Board Statement:** Not applicable.

**Informed Consent Statement:** Not applicable.

**Data Availability Statement:** Data are contained within the article.

**Conflicts of Interest:** The authors declare no conflict of interest.

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
