# Peer review of "Using Optimized Two and Three-Band Spectral Indices and Multivariate Models to Assess Some Water Quality Indicators of Qaroun Lake in Egypt"

_sustainability, doi:10.3390/su131810408_

Round 1
Reviewer 1 Report
All text should be checked for typos and standardization should be done. For example, sometimes Chl-a is written, and sometimes chl-a. Again, superscripts and subscripts in chemical formulas and symbols should be corrected. Current references should be increased proportionally.
Author Response
Reviewer # 1
We greatly appreciate your observations as well as your constructive and helpful comments. We hope that we could address your questions/comments by the explanations and revisions made in the manuscript. We believe that the manuscript is substantially improved after making the suggested revisions.
All text should be checked for typos and standardization should be done. For example, sometimes Chl-a is written, and sometimes chl-a. Again, superscripts and subscripts in chemical formulas and symbols should be corrected. Current references should be increased proportionally.
Many thanks for these observations. Symbols were checked and modified in all manuscript. Also, many references were added in all manuscript. The language of manuscript has been edited.

Reviewer 2 Report
Your work is not up to standard of this journal. Please revise it according to the following comments:
- This journal is committed to engaging with a wider public in order to promote the potential benefits cutting-edge research. Please describe in specific terms the potential impact of your work on the wider public. -
- Ideally, this article demonstrate how research results can be used in engineering practice.
- In what way does the project contribute to the SDGs? What are the trends and challenges of the technological approaches of this technique based on SDG paradigm?
You should think how transformational the research is likely to be should be made so that the outcome of the work will have an impact on the community/society facing given sustainability related challenges?
4. Write the practical applications of your work in a separate section, before the conclusions and provide your good perspectives.
5. What are the bottlenecks of this work and how did you mitigate the impacts attributed to them?
- What are the technological innovations of the work?
- What is the novelty (originality) of the work? And What is new in your work that make a difference in the body of knowledge? What has been done that goes beyond the existing research
8. How did you do quality control (QC) and quality assurance (QA) on the obtained data to validate the conclusions?
- How would this research work advance the previous work done in the existing field of study and/or across other fields?
- Based on the data obtained, what are the implications of this work (a) to the field of study, (b) to industry, (c) economy, and/or (c) to the wider public/society in general?
- How would the outcomes of work directly contribute to global climate change mitigation?
- What are the likely research impacts of this work globally, nationally and locally?
- What are the economical benefits of this work?
- This work was undertaken in laboratory scale. How to adjust and optimize the operational conditions for pilot (larger) scale?
- Make a table of comparison of this present work and other similar techniques from previously published study in terms of operational parameters and operational cost; afterwards, please give a critical analysis on its technical feasibility and applicability for upscaling this process.
16. How does the work relate, synergize or align with national contribution(s) to regional and/or international conventions, including the UN SDGs (sustainable development goals)?
17. Why do you believe your research to be important? What long-term impacts will it have on environmental protection and the wider public or the field following the completion of the project?
18. Authors must do a sufficient literature survey in this area and many progress in this research topic is largely missed.
Doi: 10.1016/j.jenvman.2021.112265
Doi: 10.1016/j.wasman.2013.10.040
19. Pls ask a native English speaker from the MDPI's proofreading service to revise and proofread your revised work before re-submission and attach the certificate of their service with your revised work.
Please respond to all of those comments in the revised manuscript by pointing out precisely and concisely on which page and in which line you have incorporated your response one by one.
Author Response
Reviewer # 2
1- This journal is committed to engaging with a wider public in order to promote the potential benefits cutting-edge research. Please describe in specific terms the potential impact of your work on the wider public. –
Response: We greatly appreciate your critical observations as well as your constructive and helpful comments. Monitoring water bodies such as lakes and rivers in terms of water quality is crucial since they are among the main sources of fresh water for various purposes (e.g. drinking, irrigation) and thus conserving water in lakes at a minimum level of pollution would be useful. It was added in introduction section from the lines 81-84.
2- Ideally, this article demonstrate how research results can be used in engineering practice.
The explanation of how research results can be used in engineering practice was written under new section (3.5. outcomes and Practical Applications of the Research), which it was suggested by reviewer.
3- In what way does the project contribute to the SDGs? What are the trends and challenges of the technological approaches of this technique based on SDG paradigm? You should think how transformational the research is likely to be should be made so that the outcome of the work will have an impact on the community/society facing given sustainability related challenges?
Response: Many thanks for this comment. Among the 17 goals of sustainable development is to have clean water everywhere worldwide. Monitoring lakes as a source of fresh water would be effective technique in water sources management. Informed decisions can be taken from the results of the assessment and monitoring lakes by means of remote sensing (ground-based and satellite-based). The explanation was written under new section (3.5. outcomes and Practical Applications of the Research).
4- Write the practical applications of your work in a separate section, before the conclusions and provide your good perspectives.
Response: Many thanks for this comment. As recommended by the reviewer a separate section on practical applications and outcomes has been added to the revised manuscript just before the conclusions (3.5. outcomes and Practical Applications of the Research) from lines 694 to 721.
- What are the bottlenecks of this work and how did you mitigate the impacts attributed to them?
Response: Many thanks for this comment. One of the main limitations to our work was to have high resolution satellite images for the study area to extrapolate the results of ground-based measurements.
6- What are the technological innovations of the work?
Response: Many thanks for this comment. The innovation here is using the remote sensing data collected by advanced instruments. These instruments can provide a substantial information locally and regionally when combined ground-base and satellite or airborne-base data for the application of detecting WQIs. The explanation was written from Line 708 to 710.
7- What is the novelty (originality) of the work? And what is new in your work that makes a difference in the body of knowledge? What has been done that goes beyond the existing research
Response: Many thanks for this comment. The novelty of our work was investigating the potential of two and three-band SRIs together with multivariate models to enhance the prediction of WQIs. Lines: 187-189
- How did you do quality control (QC) and quality assurance (QA) on the obtained data to validate the conclusions?
Many thanks. The quality control and quality assurance of the obtained data were added in the materials and methods section from line 235 to 240.
9- How would this research work advance the previous work done in the existing field of study and/or across other fields?
The potential three-band SRIs together with multivariate models to enhance the prediction of WQIs are not used in other studies. Manufacturing a simple and cost-effective three band spectral instruments can help for a primary detection of changes in lakes ecosystem. The innovation here also is using the remote sensing data collected by advanced instruments. These instruments can provide a substantial information locally and regionally when combined ground-base and satellite or airborne-base data for the application of detecting WQIs.
10- Based on the data obtained, what are the implications of this work (a) to the field of study, (b) to industry, (c) economy, and/or (c) to the wider public/society in general?
Response: We greatly appreciate your critical observations as well as your constructive and helpful comments. The main implications of this piece of research are exploring the causes of pollution in the Qaroun Lake and therefore maximize the benefits for all living organisms and even the wider public. And it was explained above.
11- How would the outcomes of work directly contribute to global climate change mitigation?
Response: Many thanks for this comment. Real assessment and monitoring of the lake by accurate and reliable techniques can help to avoid pollution hazard. Time series satellite imagery can be useful tool for managing water quality in lakes by conserving valuable natural resources. Lines 695 to 707.
12- What are the likely research impacts of this work globally, nationally and locally?
Response: Many thanks for this comment. Rural people in developing countries are at health risk from contact or even drink polluted water and therefore water quality assessment worldwide should be done precisely. Assessment of water quality parameters at local scale and then extrapolate the results at larger scale (regional and global) using remote sensing data as a robust tool in conserving life. Line: 146 to 153.
13- What are the economical benefits of this work?
Response: The chemical methods to assess monitoring and managing the WQIs in real-time and on a large scale are expensive and are limited for, and thus there is an urgent need to use reliable, cost-effective monitoring tools that can be easily deployed and assist decision-makers in assessing key indicators relevant to water quality in a comprehensive manner. Lines 139 to 143 and 717 to 719.
14- This work was undertaken in laboratory scale. How to adjust and optimize the operational conditions for pilot (larger) scale?
Response: Many thanks for this comment. In the application section, we added that a combined ground-base and satellite or airborne-based data would be a useful technique for detecting WQPs in lakes. Satellite imagery with medium and high resolution can offer better insight on ecology of lake systems. Lines: 708-710.
15- Make a table of comparison of this present work and other similar techniques from previously published study in terms of operational parameters and operational cost; afterwards, please give a critical analysis on its technical feasibility and applicability for upscaling this process.
Response: Many thanks for this comment. This suggestion need future study and it is not one of our targets in this study. Our methods are cheaper compare to chemical analysis.
16- How does the work relate, synergize or align with national contribution(s) to regional and/or international conventions, including the UN SDGs (sustainable development goals)?
Response: Many thanks for this comment. As a result of water scarcity in arid and semi-arid regions the outcomes of this research can help decision maker managing water budgets properly. For example, the challenge of facing shortage of fresh water in a country like Egypt needs this sort of new technologies (remote sensing) as well as using low quality water in industrial and agricultural fields to minimize hunger which is one of the SDGs. Lines: 710 -716.
17- Why do you believe your research to be important? What long-term impacts will it have on environmental protection and the wider public or the field following the completion of the project?
Response: Many thanks for this comment. What makes this research important is that multivariate models (e.g. PLSR, ) was effective in enhancing the detection of WQPs. Two-band and three band ratios derived from ground-based spectral data can help identifying the optimum indices that can be employed when using satellite images with low number of spectral bands.
18- Authors must do a sufficient literature survey in this area and many progress in this research topic is largely missed.
Doi: 10.1016/j.jenvman.2021.112265
Doi: 10.1016/j.wasman.2013.10.040
Many thanks. Many literature reviews were added in the work under the introduction section in page 2 from line 62 to line 71.
19- Pls ask a native English speaker from the MDPI's proofreading service to revise and proofread your revised work before re-submission and attach the certificate of their service with your revised work.
Response: Many thanks for this comment. The English of manuscript has been edited.

Reviewer 3 Report
The authors proposed and evaluated a method to interpret water quality parameters in a specific lake using spectral reflectance indices. Concrete data are shown with careful discussion. However, the authors need to discuss whether this method is promising to be used in other circumstances and why. Then the authors need to make practical recommendations about how this method can be applied. In the current version, the authors only focused on the specific lake. To publish the work in an international journal, this is not enough. In addition, after showing the results, the authors need to make a comparison with previous studies and then make necessary discussion mechanistically. Please see additional specific comments shown below.
Specific comments:
Line 40: What are two-band and three-band? Please define them.
Lines 42 to 43: I think this sentence can be removed. You did not investigate how salinity changes influenced the ecosystems.
Line 45: Quantitatively indicate the moderate and strong relationships.
Line 49: Double check the numbers. There are two items NSRIs-3b had relationships with, but three R2 values.
Line 72: When did it change to a salt lake?
Line 107: Specify how to keep minimum effects.
Lines 109 to 110: Give background information about spectral analysis here.
Lines 118 to 120, “Although these measurements are accurate…”: This sentence sounds weird. You just indicated methods are limited and we need to develop better methods. Why did you say the measurements are accurate?
Lines 128 to 132: You need to expand the discussion. Please review literature and show what previous studies reported in terms of all water quality parameters you measured in this study. You measured five WQIs. You need to show whether there are spectral indices for each WQI.
Lines 137 to 138: Explain what are the two-band and three-band spectral indices. You need to discuss what previous studies have done and what has to be investigated further.
Line 146, “PLSR”: When the abbreviation is shown for the first time, you need to give the full name and explain the background information. Please indicate what is PLSR and how to use it.
Lines 150 to 151, “the PLSR is mainly powerful in cases…”: Why?
Line 153: What is VIS-SWIR?
Line 157, “SRIs (commonly used SRIs, NSRIs-2b and NSRIs-3b)”: Give background information about these types.
Line 166: Is this research useful to other lakes?
Line 230: Are the spectra data influenced by the season?
Line 235: Please indicate first how 2-D and 3-D SRIs were established.
Line 236, “302 nm to 1148 nm”: Explain why you chose this range.
Lines 238 to 240: Is this method to determine SRIs site-specific? Can these SRIs be used in other places?
Line 244: What are R1 and R2?
Line 249: How did you establish this equation? Please explain the mechanisms. Did you follow previous studies? If yes, please cite the references.
Line 263: What is RMSE?
Line 281, Section 3.1: I think this part should be condensed. This cannot be considered as one of the main achievements in this study. To analyze water quality in a specific lake should be not the main content of a research paper. The analytical data should serve as the basic information used to establish the integrated approach you proposed. In addition, as shown in my comments, your discussion after you showed the results was not in-depth. If you would like to discuss the reasons leading to such results, you need to take more samples and make in-depth investigation.
Line 292: Which results showed the steep temperature gradients?
Line 295, “…which demonstrated an increase in photosynthetic activity of planktonic algae”: Specify how you got this conclusion.
Line 310: How did you draw the map? Did you only use the measurements obtained in this study?
Line 326: Did you investigate what species were present in the water? It is better to identify them. Then discuss the water quality that these species can live with.
Line 341: You can take samples to identify TP levels in wastewater and farm effluent.
Line 354, Table 4: Please make the table better to be understood, especially for the second column. Aquatic life with different numbers is confusing. What was your purpose to show this table? How can you solve the problems reflected in the table?
Lines 360 to 362: Show references and specify how well the indices estimated water quality parameters in previous studies.
Lines 367 to 269: Show the results of quantitative analyses.
Lines 376 to 378, “There were clear differences of SRIs…”: Please explain the reason in more detail.
Line 380, Table 5: Please explain how to read these numbers. What do a, b, c, etc. mean?
Line 387: Should be WQIs.
Figure 4: Indicate what the colours mean.
Figure 5: Indicate what the colours mean.
Line 407: Please explain how to read these plots. How can you find WQIs in these plots? Where are the hotspots?
Line 411: What are published and newly selected SRIs? You have not introduced them.
Line 420: Do you mean “linear regression models”?
Lines 423 to 426: Please define the ranges of R2 values for moderate and strong relationships.
Lines 431 to 432, “It seems that the spectral index…”: Please discuss the reason.
Lines 433 to 444: You discussed some studies. Please discuss how their results, especially R2 values, compare to your results.
Line 444: When you discuss previous studies, please make sure to compare their results to yours and indicate which one is better associated with reasons.
Line 454: It is better to start a new paragraph to discuss TP.
Lines 464 to 466: Somewhere in the article, please specify how high or low a R2 value stands for a good or weak relationship. In addition, please discuss why TDS was different from others.
Line 470: Discuss why NSRIs-3b usually gives a better estimation. You need quantitative discussion to demonstrate why this approach is better.
Lines 481 to 482: Provide more discussion about this statement. It is not clear and seems contradictory.
Lines 490 to 493: Please explain why NSRIs-3b had the best performance.
Lines 507 to 509: Please expand the discussion. Theoretically, if one waveband could reflect a water parameter sensitively and accurately, why do we need to consider other wavebands? Please discuss the mechanisms.
Lines 517 to 520: Compare Xing's method to your method and discuss which one is better. In general, after you review previous studies, you need to discuss what you have improved in this study.
Table 6: What do the asterisks mean for R2 values?
Author Response
Reviewer # 3
Response: We greatly appreciate your critical observations as well as your constructive and helpful comments. We hope that we could address your questions/comments by the explanations and revisions made in the manuscript. We believe that the manuscript is substantially improved after making the suggested revisions.
The authors proposed and evaluated a method to interpret water quality parameters in a specific lake using spectral reflectance indices. Concrete data are shown with careful discussion. However, the authors need to discuss whether this method is promising to be used in other circumstances and why. Then the authors need to make practical recommendations about how this method can be applied. In the current version, the authors only focused on the specific lake. To publish the work in an international journal, this is not enough. In addition, after showing the results, the authors need to make a comparison with previous studies and then make necessary discussion mechanistically. Please see additional specific comments shown below.
Response: Many thanks for these comments. According to your general and specific comments, the manuscript was improved. We addressed your questions/comments by the explanations and revisions made in the manuscript.
Line 40: What are two-band and three-band? Please define them.
Response: Many thanks for this comment. Two-band and three-band were defined in abstract. Lines: 41-42
Lines 42 to 43: I think this sentence can be removed. You did not investigate how salinity changes influenced the ecosystems.
Response: Many thanks for this comment. The sentence (progressive increase in salinity accelerates the degradation of the lake's aquatic ecosystem) was removed.
Line 45: Quantitatively indicate the moderate and strong relationships.
Response: Many thanks for this comment. The modification was added.
Line 49: Double check the numbers. There are two items NSRIs-3b had relationships with, but three R2 values.
Response: Many thanks for this comment. Yes, the missing water indicator is TSS and it was added. Line: 47.
Line 72: When did it change to a salt lake?
Many thanks for this construction. Investigation of salinity changes that influenced the ecosystems was added under the introduction section from line 87 to 95.
Line 107: Specify how to keep minimum effects.
Many thanks for this construction. These words were deleted from the manuscript.
Lines 109 to 110: Give background information about spectral analysis here.
Response: Many thanks for this comment. Spectral analysis was removed from Lines 133 and we kept the information only about GIS to be clear for the reader. Information about spectral measurements can be observed from the line 136 (Ground based-remote sensing based on spectral reflectance…….).
Lines 118 to 120, “Although these measurements are accurate…”: This sentence sounds weird. You just indicated methods are limited and we need to develop better methods. Why did you say the measurements are accurate?
Response: Many thanks for this comment. I mean the laboratory analysis is accurate but at the same time they are expensive, providing no real-time spatial and temporal changes in water quality. For that we need rapidly methods to overcome this problem. To prevent confusion, the sentence was removed.
Lines 128 to 132: You need to expand the discussion. Please review literature and show what previous studies reported in terms of all water quality parameters you measured in this study. You measured five WQIs. You need to show whether there are spectral indices for each WQI.
Response: Many thanks for this comment. Other review literature related WQIs was added. Lines: 153-161.
Lines 137 to 138: Explain what are the two-band and three-band spectral indices. You need to discuss what previous studies have done and what has to be investigated further.
Response: Many thanks for this comment. Through the improvement in manuscript which suggest by the reviewer, we point to the two-band and three-band spectral indices in sections of introduction, results and discussion.
Line 146, “PLSR”: When the abbreviation is shown for the first time, you need to give the full name and explain the background information. Please indicate what is PLSR and how to use it.
Response: Many thanks for this comment. PLSR was defined. Line: 175.
Lines 150 to 151, “the PLSR is mainly powerful in cases…”: Why?
Response: Many thanks for this comment. The PLSR is mainly powerful in cases of spectral analysis. (1) PLSR has been proposed to resolve the strong multi-collinear and noisy variables. As well as (2) the number of latent factors (ONLFs) was determined using (LOOCV), and the best ONLFs are that yielding the greatest R2 and the smallest RMSE in order to represent the calibration data with no over-fitting or under-fitting as indicate under section 2.6 of Material and Methods
Line 153: What is VIS-SWIR?
Response: Many thanks for this comment. The definition of VIS-SWIR was added. Line: 183.
Line 157, “SRIs (commonly used SRIs, NSRIs-2b and NSRIs-3b)”: Give background information about these types.
Response: Many thanks for this comment. Background information about these types was added.
Line 166: Is this research useful to other lakes?
Response: Many thanks for this comment. As we wrote in conclusion section from the line 745 to 747. In future studies, we will test the findings of spectral indices algorithms and PLSR models based on spectral bands and spectral indices to test stability of them the under different conditions of Lakes and Rivers.
Line 230: Are the spectra data influenced by the season?
Response: Many thanks for this comment. In general, the summer in Egypt, the amount of sun radiation around noon is more stable in two seasons. As well as, to keep fluctuation at a minimum and to reduce the impact of changes in sun zenith angle, spectra were taken around noon time. In addition, there was no clear difference in changes in water quality indicators in both seasons as presented in Figure 2 and 3. For that there were no clear influenced by the season.
Line 235: Please indicate first how 2-D and 3-D SRIs were established.
Response: Many thanks for this comment. The established of 2-D and 3-D SRIs were explained in details under section 2.5. .
Line 236, “302 nm to 1148 nm”: Explain why you chose this range.
Response: Many thanks for this comment. We added the explanation about the range of spectrometer in the text in 284. As well as other reason for that our spectrometer (tec5 AG, Oberursel, Germany) had spectral range of 302 and 1148 nm.
Lines 238 to 240: Is this method to determine SRIs site-specific? Can these SRIs be used in other places?
The methods for extract SRIs using 2-D and 3-D correlogram maps are constant. But in this study, SRIs site-specific was extracted from water spectral reflectance of Qaroun Lakes. In future studies, we will test the findings of spectral indices algorithms and PLSR models based on spectral bands and spectral indices to test stability of them the under different conditions of Lakes and Rivers.
Line 244: What are R1 and R2?
Response: Many thanks for this comment. R1 and R2 are the values of spectral reflectance at selected wavelengths. Lines: 290 and 294.
Line 249: How did you establish this equation? Please explain the mechanisms. Did you follow previous studies? If yes, please cite the references.
Response: Many thanks for this comment. The cite the reference was added.
Line 263: What is RMSE?
Response: Many thanks for this comment. RMSE is root mean square error. The definition of RMSE was added in the text. Line: 326.
Line 281, Section 3.1: I think this part should be condensed. This cannot be considered as one of the main achievements in this study. To analyze water quality in a specific lake should be not the main content of a research paper. The analytical data should serve as the basic information used to establish the integrated approach you proposed. In addition, as shown in my comments, your discussion after you showed the results was not in-depth. If you would like to discuss the reasons leading to such results, you need to take more samples and make in-depth investigation.
Response: Many thanks for this comment. We presented by this way to give the state of surface water quality of Qaroun Lake as one of the objective of this study.
Line 292: Which results showed the steep temperature gradients?
Many thanks. These words were deleted from the sentence under the results and discussion from line 354 to line 3655.
Line 295, “…which demonstrated an increase in photosynthetic activity of planktonic algae”: Specify how you got this conclusion.
Many thanks. Fluctuations of pH levels could be the result of the phytoplankton photosynthetic status since; pH value is strongly regulated by the rate of CO2 consumption through the phytoplankton photosynthetic activities (El-Otify and Iskaros 2015; Khalifa 2014). This paragraph was added in the manuscript under the results and discussion section in from line 361 to line 364.
Line 310: How did you draw the map? Did you only use the measurements obtained in this study?
Many thanks. The map was drawn using ArcGIS software according to the obtained measuring results.
Line 326: Did you investigate what species were present in the water? It is better to identify them. Then discuss the water quality that these species can live with.
Many thanks. It was discussed under results and discussion section from line 395 to line 405.
Line 341: You can take samples to identify TP levels in wastewater and farm effluent.
Not available in this time.
Line 354, Table 4: Please make the table better to be understood, especially for the second column. Aquatic life with different numbers is confusing. What was your purpose to show this table? How can you solve the problems reflected in the table?
Many thanks. This table presented only classification of surface water for suitability to aquatic life in Qaroun Lake according to water quality parameters over two years
Lines 360 to 362: Show references and specify how well the indices estimated water quality parameters in previous studies.
Response: Many thanks for this comment. The references were added in introduction section and the explanation of how well the indices estimated water quality parameters was explained in section of results and discussion.
Lines 367 to 369: Show the results of quantitative analyses.
Response: Many thanks for this comment. Example of quantitative analyses was added from the line 452 to 455.
Lines 376 to 378, “There were clear differences of SRIs…”: Please explain the reason in more detail.
Response: Many thanks for this comment. Explanation of the reason was added from the line 462 to 469.
Line 380, Table 5: Please explain how to read these numbers. What do a, b, c, etc. mean?
Response: Many thanks for this comment. The explanation was added under the table.
Line 387: Should be WQIs.
Response: Many thanks for this comment. It was corrected to WQIs. Line: 483.
Figure 4: Indicate what the colours mean.
Response: Many thanks for this comment. The colours mean the values of R2. For that figure 4 was modified by adding R2 above legend of scale colours.
Figure 5: Indicate what the colours mean.
Response: Many thanks for this comment. The colours mean the values of R2. For that figure 5 was modified by adding R2 above legend of scale colours.
Line 407: Please explain how to read these plots. How can you find WQIs in these plots? Where are the hotspots?
Response: Many thanks for this comment. This paragraph was modified to be clear for the reader. The hotspot areas based on colour scale for the best R2 identify the best relationships between SRIs and WQIs. Then the NSRIs of 2-D and 3-D correlogram maps were selected based on highest R2. Hotspot is related to colour scales which was used to detect the best R2 between SRI and each WQI.
Line 411: What are published and newly selected SRIs? You have not introduced them.
Response: Many thanks for this comment. Published and newly selected SRIs were replaced by commonly used SRIs, NSRIs-2b and NSRIs-3b to be clear for the reader. Line: 514.
Line 420: Do you mean “linear regression models”?
Response: Many thanks for this comment. Yes, linear regression models. It was corrected in the text. Line: 525.
Lines 423 to 426: Please define the ranges of R2 values for moderate and strong relationships.
Response: Many thanks for this comment. The ranges of R2 values for moderate and strong relationships were added in the text from the line 529 to 532.
Lines 431 to 432, “It seems that the spectral index…”: Please discuss the reason.
Response: Many thanks for this comment. The reason of that was added in the text from the lines 538 to 540 as well as other studies was added to confirm our results from lines 540 to 558.
Lines 433 to 444: You discussed some studies. Please discuss how their results, especially R2 values, compare to your results.
Response: Many thanks for this comment. R2 values for each previous study were added in the text from line 540 to 558.
Line 444: When you discuss previous studies, please make sure to compare their results to yours and indicate which one is better associated with reasons.
Response: Many thanks for this comment. From 533 to 437, we presented the best SRI was found in this study and also through the Figures 4, 5 and 6 can show more details about the relationships between SRIs and WQIs, then from the line 537 to 558. the results of previous studies were presented under different conditions.
Line 454: It is better to start a new paragraph to discuss TP.
Response: Many thanks for this comment. It was done.
Lines 464 to 466: Somewhere in the article, please specify how high or low a R2 value stands for a good or weak relationship. In addition, please discuss why TDS was different from others.
Response: Many thanks for this comment. The sentence was modified to be clear for the reader and good or weak relationship was indicated in the text As well as the reason of the TDS was different from others was discussed from line 580 to 588.
Line 470: Discuss why NSRIs-3b usually gives a better estimation. You need quantitative discussion to demonstrate why this approach is better.
Response: Many thanks for this comment. Discuss why NSRIs-3b usually gives a better estimation was added in the text from lines 519 to 526.
Lines 481 to 482: Provide more discussion about this statement. It is not clear and seems contradictory.
Response: Many thanks for this comment. The sentence was improved to be clear for the reader.
Lines 490 to 493: Please explain why NSRIs-3b had the best performance.
Response: Many thanks for this comment. Discuss why NSRIs-3b usually gives a better estimation was added in the text from lines 555 to 561.
Lines 507 to 509: Please expand the discussion. Theoretically, if one waveband could reflect a water parameter sensitively and accurately, why do we need to consider other wavebands? Please discuss the mechanisms.
Response: Many thanks for this comment. The comment was discussed above in the text from lines 594 to 605.
Lines 517 to 520: Compare Xing's method to your method and discuss which one is better. In general, after you review previous studies, you need to discuss what you have improved in this study.
Response: Many thanks for this comment. The explanation of this comment was added in the text from the lines 660 to 673.
Table 6: What do the asterisks mean for R2 values?
Response: Many thanks for this comment. The explanation of asterisks were added under the table 6.

Reviewer 4 Report
The section 2.5 is not easy to read. I would appreciate it if you make it more readable. I would also like to know the spectral resolution and if this way of identifying WQIs was used in other works. Why for example is NSI calculated like given in eq.(2)? What is the rationale? The next section, 2.6 is written very nicely, it was easy to understand.
I do not understand the values given in Table 5. To me, those numbers and letters do not mean anything. Ok, they show that SRIs are different at 16 locations, but ..this could also be a totally random result. Is this the only way to represent the data?
While I can follow the results and how you ended up with conclusion, what bothers me is that you start with indices that others were using, then try to obtain better results by introducing some new and those which include three bands. Then only, you use PLSR to find the relationship between the spectral indices and water quality parameters such as TDS etc. If you start with PLSR using ALL wavelengths and try to model directly TDS, Transparency, TSS and others, you could inspect regression vectors for influential variables, or use some of the many variable selection methods to identify which of the variables (wavelengths) are most effective. This way, you do not rely on the previous experience of people who might have been wrong, and you find the exact bands that give you the optimized combination of wavelengths that you can use. I think this is much better idea. Could you please comment on this? Why did you do it this way?
Author Response
Reviewer # 4
Response: We greatly appreciate your critical observations as well as your constructive and helpful comments. We hope that we could address your questions/comments by the explanations and revisions made in the manuscript. We believe that the manuscript is substantially improved after making the suggested revisions. As well as the all manuscript was improved and discussion was supported.
The section 2.5 is not easy to read. I would appreciate it if you make it more readable.
Response: Many thanks for this comment. The section 2.5 has been rewritten to be easy for the reader.
I would also like to know the spectral resolution and if this way of identifying WQIs was used in other works.
Response: Many thanks for this comment. Spectral resolution of spectrometer is 2 nm. It was written under section (2.4. Ground-Based Remote-Sensing Measurements). Line: 261.
Why for example is NSI calculated like given in eq.(2)? What is the rationale?
Response: Many thanks for this comment. NSI is considered one of the most important mathematical equations that are used in calculating the spectral indices and it gave the best results than the ratio ratio spectral index in calculating the relationship between three band spectral indices and water quality indicators.
The next section, 2.6 is written very nicely, it was easy to understand.
Response: Thank you very much for this moral support.
I do not understand the values given in Table 5. To me, those numbers and letters do not mean anything. Ok, they show that SRIs are different at 16 locations, but ..this could also be a totally random result. Is this the only way to represent the data?
Response: Many thanks for this comment. This method is good to present the significant differences for each spectral reflectance indices at different 16 locations. a, b, c, etc. mean the same letters are not significantly different from one another based on Duncan’s test at a p ≤ 0.05 significance level. And this sentence was added under the Table 5. This method is only easy way to present the significant differences between several indices. In this study, we tested 22 SRIs and it is better to included them in Table by this way.
While I can follow the results and how you ended up with conclusion, what bothers me is that you start with indices that others were using, then try to obtain better results by introducing some new and those which include three bands. Then only, you use PLSR to find the relationship between the spectral indices and water quality parameters such as TDS etc. If you start with PLSR using ALL wavelengths and try to model directly TDS, Transparency, TSS and others, you could inspect regression vectors for influential variables, or use some of the many variable selection methods to identify which of the variables (wavelengths) are most effective. This way, you do not rely on the previous experience of people who might have been wrong, and you find the exact bands that give you the optimized combination of wavelengths that you can use. I think this is much better idea. Could you please comment on this? Why did you do it this way?
Response: Many thanks for this comment. In the beginning we tested your suggest method by training and validating PLSR models using all wavelengths to predict the TDS, Transparency, TSS, Chl-a and TP, but we found the results is not good compare to PLSR model based on NSRIs-2b and NRSIs-3b. PLSR models using all wavelengths and inspect regression vectors for influential variables, or use some of the many variable selection methods to identify which of the variables (wavelengths), we have used it in our pervious published paper such as (doi: 10.3389/fpls.2019.01537).
Please find the table results of calibration and validation models based on the wavelengths
ONLFs |
Measured variables |
Calibration |
Validation |
||||
Equation |
R2cal |
RMSEC |
Equation |
R2val |
RMSEV |
||
3 |
TDS |
y = 0.5265x + 16893 |
0.53*** |
1693 |
y = 0.3426x + 23492 |
0.29** |
2098 |
2 |
Transparency |
y = 0.4631x + 35.401 |
0.46*** |
21.52 |
y = 0.4267x + 37.763 |
0.38** |
23.03 |
4 |
TSS |
y = 0.7101x + 10.554 |
0.71*** |
9.09 |
y = 0.6199x + 13.886 |
0.47** |
12.84 |
2 |
Chl-a |
y = 0.5722x + 0.0391 |
0.57*** |
0.023 |
y = 0.5353x + 0.0427 |
0.49** |
0.04 |
2 |
TP |
y = 0.515x + 0.1703 |
0.52*** |
0.14 |
y = 0.4739x + 0.1853 |
0.42** |
0.16 |

Round 2
Reviewer 2 Report
Well done!
Author Response
Well done
Response: We greatly appreciate your critical observations as well as your constructive and helpful comments.

Reviewer 3 Report
I appreciate the authors’ responses to my comments. I can see the manuscript has been revised. However, I found some comments have not been addressed well. Please see my follow-up comments shown below.
Specific comments:
Response to Line 45: I mean please add quantitative data to show moderate and strong relationships.
Response to Line 157: I may not express my meaning clearly. I mean please explain what are NSRIs-2b and NSRIs-3b. Why do you name them 2b and 3b?
Response to Line 166: A better expression could be "In the future, the method proposed in this study combining spectral indices algorithms and PLSR models could be evaluated further to improve its stability under various conditions of rivers and lakes.”
Response to Line 235: In the revised manuscript, on Line 271, the three-band should be NSRIs-3b.
Response to Line 354, Table 4: You show "aquatic life" as the title of the second column with many different numbers underneath. This is confusing and hard to be understood. Please figure it out.
Response to Lines 464 to 466: I think the revision is now from Line 564 to 574. I can understand your meaning but it is hard to read. Please improve the writing.
Response to Line 470: I think your explanation is that the red region is more sensitive to the changes in Chl-a. Why? Can you show references to support this statement?
Response to Lines 481 to 482: The writing could be "using several wavebands sensitive to water quality parameters through SRIs combined with PLSR models could enhance the performance of the models to predict the WQIs."
Responses to Lines 507 to 509: Please improve the writing from Line 595 to Line 601.
Responses to Lines 517 to 520: I am not sure if the revision corresponding to this response is shown on Lines 625 to 638. The writing is not good. I do not understand it. It seems that the revision does not address my comment. I hope you can make a comparison between your method with others and discuss which one is the best.
Author Response
Reviewer # 3
Response: We greatly appreciate your critical observations as well as your constructive and helpful comments. We hope that we could address your questions/comments by the explanations and revisions made in the manuscript. We believe that the manuscript is substantially improved after making the suggested revisions.
Specific comments:
Response to Line 45: I mean please add quantitative data to show moderate and strong relationships.
Response: Many thanks for this comment. Quantitative data was added in Lines 46,47 and 48.
Response to Line 157: I may not express my meaning clearly. I mean please explain what are NSRIs-2b and NSRIs-3b. Why do you name them 2b and 3b?
Response: Many thanks for this comment. 2b is two b and 3b is three band and they was added in 168.
Response to Line 166: A better expression could be "In the future, the method proposed in this study combining spectral indices algorithms and PLSR models could be evaluated further to improve its stability under various conditions of rivers and lakes.”
Response: Many thanks for this comment. The sentence was modified according to reviewer suggestion.
Response to Line 235: In the revised manuscript, on Line 271, the three-band should be NSRIs-3b.
Response: Many thanks for this comment. It was modified.
Response to Line 354, Table 4: You show "aquatic life" as the title of the second column with many different numbers underneath. This is confusing and hard to be understood. Please figure it out.
Response: Many thanks for this comment. Different numbers are related to the standard of aquatic life and it was written in the table. The title of the table was modified to be clear for the reader and also [71] reference was added and (-) was explained under the table.
Response to Lines 464 to 466: I think the revision is now from Line 564 to 574. I can understand your meaning but it is hard to read. Please improve the writing.
Response: Many thanks for this comment. It was improved from line 597 to 608.
Response to Line 470: I think your explanation is that the red region is more sensitive to the changes in Chl-a. Why? Can you show references to support this statement?
Response: Many thanks for this comment. We added three references including the bands from red region from line 542 to 556.
Response to Lines 481 to 482: The writing could be "using several wavebands sensitive to water quality parameters through SRIs combined with PLSR models could enhance the performance of the models to predict the WQIs."
Response: Many thanks for this comment. The sentence was added from line 618 to 620.
Responses to Lines 507 to 509: Please improve the writing from Line 595 to Line 601.
Response: Many thanks for this comment. The writing was improved from the line 632 to 637.
Responses to Lines 517 to 520: I am not sure if the revision corresponding to this response is shown on Lines 625 to 638. The writing is not good. I do not understand it. It seems that the revision does not address my comment. I hope you can make a comparison between your methods with others and discuss which one is the best.
Many thanks for your comments. Through the section of the results and discussion we have presented the quantitative data of our methods and as well as for previous studies for spectral indices and PLSR methods. For example; for spectral indices was presented from line 523 to 583.

Round 3
Reviewer 3 Report
The manuscript is better than before.